EMBO
Molecular Medicine

# Methotrimeprazine is a neuroprotective antiviral in JEV infection via adaptive ER stress and autophagy

Surendra K Prajapat[1], Laxmi Mishra[1], Sakshi Khera[1], Shadrack D Owusu[1,6], Kriti Ahuja[2], Puja Sharma[1], Eira Choudhary[1], Simran Chhabra[1], Niraj Kumar[3], Rajan Singh [iD][4,7], Prem S Kaushal [iD][3], Dinesh Mahajan [iD][5], Arup Banerjee[1], Rajender K Motiani [iD][2], Sudhanshu Vrati[1] & Manjula Kalia [iD][1✉]

## Abstract

Japanese encephalitis virus (JEV) pathogenesis is driven by a combination of neuronal death and neuroinflammation. We tested 42 FDA-approved drugs that were shown to induce autophagy for antiviral effects. Four drugs were tested in the JE mouse model based on in vitro protective effects on neuronal cell death, inhibition of viral replication, and anti-inflammatory effects. The antipsychotic phenothiazines Methotrimeprazine (MTP) & Trifluoperazine showed a significant survival benefit with reduced virus titers in the brain, prevention of BBB breach, and inhibition of neuroinflammation. Both drugs were potent mTOR-independent autophagy flux inducers. MTP inhibited SERCA channel functioning, and induced an adaptive ER stress response in diverse cell types. Pharmacological rescue of ER stress blocked autophagy and antiviral effect. MTP did not alter translation of viral RNA, but exerted autophagy-dependent antiviral effect by inhibiting JEV replication complexes. Drug-induced autophagy resulted in reduced NLRP3 protein levels, and attenuation of inflammatory cytokine/chemokine release from infected microglial cells. Our study suggests that MTP exerts a combined antiviral and anti-inflammatory effect in JEV infection, and has therapeutic potential for JE treatment.

**Keywords** Antipsychotic; Microglia; Neuroinflammation; Phenothiazines; Trifluoperazine
**Subject Categories** Microbiology, Virology & Host Pathogen Interaction; Neuroscience; Pharmacology & Drug Discovery

## Introduction

Japanese encephalitis virus (JEV) belongs to the *Flaviviridae* family that includes several pathogenic arboviruses such as West Nile virus (WNV), Yellow fever virus and Dengue virus (DENV). JEV is transmitted by infected *Culex* mosquitoes and is maintained through an enzootic cycle between birds, pigs and other vertebrate hosts. The disease is endemic in south-east Asian countries including India, with both epidemic and sporadic occurrences (Sharma et al, 2021). Over the years the virus has shown significant geographical expansion into regions not previously reported to have JE (Mulvey et al, 2021).

JEV is neurotropic, and its clinical manifestations range from mild febrile illness to encephalitis and death (Sips et al, 2012). The pediatric population is most severely affected, and treatment is mostly supportive with no effective antiviral therapy available (Turtle and Solomon, 2018). Virus induced perivascular and central nervous system (CNS) inflammation is linked to elevated intracranial pressure, seizures, movement disorders and flaccid paralysis (Salimi et al, 2016; Sharma et al, 2021; Solomon et al, 2002). Neuronal damage to the thalamus and brain stem often results in permanent neurological sequelae among the survivors (Misra and Kalita, 2010).

Following inoculation through a mosquito bite, the virus first replicates in the local dermal cells such as fibroblasts, endothelial cells and tissue-resident dendritic cells (DCs), and spreads to local lymph nodes and other organs (Filgueira and Lannes, 2019; Sharma et al, 2021). The virus also replicates in monocytes/macrophages and in most cases, is cleared by an effective peripheral immune response (Aleyas et al, 2009; Chauhan et al, 2021; Choi et al, 2019). The virus can invade the CNS either through basolateral release from infected brain microvascular endothelial cells (BMECs), or diapedesis of infected peripheral immune cells. JEV replicates efficiently in neurons, microglia and astrocytes. Production of inflammatory cytokines and metalloproteases by JEV infected BMECs, microglia, and astrocytes triggers the degradation of tight junction proteins leading to loss of brain endothelial barrier permeability. Studies have shown that blood-brain barrier (BBB) breach is not a cause, but a consequence of virus infection of the CNS and neuroinflammation (Li et al, 2015). Neuronal cell death, which is augmented by neuroinflammation is the major driver of pathogenesis (Chen et al, 2010).

[1]Virology Research Group, Regional Centre for Biotechnology, NCR Biotech Science Cluster, Faridabad 121001, India. [2]Laboratory of Calciomics and Systemic Pathophysiology, Regional Centre for Biotechnology, NCR Biotech Science Cluster, Faridabad 121001, India. [3]Structural Biology & Translation Regulation Laboratory, Regional Centre for Biotechnology, NCR Biotech Science Cluster, Faridabad 121001, India. [4]Advanced Technology Platform Centre, Regional Centre for Biotechnology, NCR Biotech Science Cluster, Faridabad 121001, India. [5]Chemistry and Drug Design Lab, Centre for Drug Design and Discovery, Translational Health Science and Technology Institute, NCR Biotech Science Cluster, Faridabad 121001, India. [6]Present address: Institut de Biologie Moléculaire et Cellulaire (IBMC), Université de Strasbourg, 67000 Strasbourg, France. [7]Present address: Department of Life Sciences, Shiv Nadar University, Greater Noida 201314, India. ✉E-mail: manjula@rcb.res.in

JEV being an RNA virus, replicates in close association with ER derived membranes, and results in the activation of stress responses such as the unfolded protein response (UPR), ER stress, generation of ROS, and upregulation of autophagy (Sharma et al, 2014; Sharma et al, 2017; Sharma et al, 2018; Su et al, 2002; Yu et al, 2006). In the context of JEV, cellular autophagy is upregulated through the activation of ER and oxidative stress, and functions primarily as an antiviral mechanism by restricting virus replication and cell death (Sharma et al, 2014; Sharma et al, 2017; Sharma et al, 2018). At later time points of infection, autophagy dysregulation is observed, which enhances virus induced neuronal death. This lead us to hypothesize that autophagy upregulation could inhibit virus replication, neuronal cell death and neuroinflammation, and is thus likely to be neuroprotective.

Established defects in autophagy in conditions such as cancer, neurodegeneration, inflammation and metabolic disorders have lead researchers to focus on the discovery of novel drugs/compounds that can modulate autophagy. Autophagy upregulation has also been shown to have therapeutic potential for neurodegenerative diseases (Park et al, 2020; Rubinsztein et al, 2012). Several FDA-approved drugs have been shown to enhance autophagy. Therefore, they have the potential to be repurposed in disease conditions where autophagy upregulation is likely to be beneficial.

Here we have examined FDA-approved drugs with autophagy inducing potential for their effect on JEV infection in vitro and in mouse model of disease. The typical antipsychotic drugs of the phenothiazine family Methotrimeprazine (MTP) and Trifluoperazine (TFP) showed strong inhibition of virus replication, and microglial/astrocyte inflammation, along with significant protection in the mouse model of disease. These drugs induced mechanistic target of rapamycin (mTOR)-independent functional autophagy flux, and their antiviral and anti-inflammatory activity was observed to be autophagy dependent. MTP treatment resulted in ER calcium dysregulation, low eukaryotic translation initiation factor 2A (eIF2α) phosphorylation and a unique adaptive ER stress gene signature. Our study suggests that MTP induces adaptive ER stress and autophagy in diverse cell types creating an antiviral and neuroprotective environment during JEV infection.

# Results

## Primary screening of autophagy inducing FDA approved drugs for antiviral and anti-inflammatory effects in vitro

Studies from our laboratory have shown that JEV infection induced cellular stress responses such as ER and oxidative stress, result in the activation of the UPR and autophagy, that play crucial roles in regulating JEV replication and cell death (Sharma et al, 2014; Sharma et al, 2017; Sharma et al, 2018). Since autophagy upregulation has potential neuroprotective roles, we tested a panel of FDA-approved drugs that have been documented as autophagy-inducers, for any anti-JEV effect. The study was initiated with forty-two drugs (Appendix Table S1), and all were tested at a concentration of 10 μM which was established as non-cytotoxic in the mouse Neuro2a cell line (Appendix Fig. S1A). A primary screening was performed using virus induced neuronal cell death assay. JEV infection of Neuro2a cells results in MOI and time-dependent cell death. A 5 MOI infection for 48 h, which results in

~80% cell death was chosen for the assay (Appendix Fig. S1B). From the panel ten drugs: Bromhexine, Clonidine, Flubendazole, Fluoxetine, Lithium chloride, Memantine, Metformin, MTP, Rilmenidine, and Sodium valproate showed reduction in JEV-induced cell death (Appendix Fig. S1C), and were short-listed for further studies. All drugs resulted in a significant reduction in JEV RNA levels (Fig. 1A), and four drugs: Clonidine, Fluoxetine, Memantine and MTP significantly reduced virus titers in Neuro2a cells (Fig. 1B).

As previously reported (Chen et al, 2010), we observed that JEV infects microglial cells efficiently (Appendix Fig. S2A) and results in robust secretion of proinflammatory cytokines such as IL-6, RANTES, TNF-α, and MCP-1 (Appendix Fig. S2B–E). After establishing non-cytotoxic concentrations (Appendix Fig. S2F), the ten short-listed drugs were checked for their effect on JEV replication (Appendix Fig. S2G), and inhibition of proinflammatory cytokine release from infected N9 microglial cells (Fig. 1C–F).

Based on observations from both neuronal and microglial cells, five drugs appeared as promising antivirals: Flubendazole, Fluoxetine, Memantine, MTP and Rilmenidine. These drugs were further tested for their effect on JEV protein translation/replication complex formation (Fig. 1G,H), and ROS production (Fig. 1I) in Neuro2a cells and a significant inhibition was observed.

## Phenothiazines exert an antiviral and anti-inflammatory effect and show protection in JEV mouse model

We next tested Flubendazole, Fluoxetine, MTP and Rilmenidine in a JEV infected C57BL/6 mouse survival assay (Appendix Fig. S3). A mouse-adapted isolate JEV-S3 was used, which results in development of typical encephalitis symptoms: loss of weight, body stiffening, piloerection, hind limb paralysis by 5–6 days post-infection (dpi), and death within 2–3 days of symptom onset (Tripathi et al, 2021). The JEV infected mice developed typical disease symptoms and showed a median survival time (MST) of 8–9 days (Fig. 2A,B; Appendix Fig. S3). Brain viremia was detectable by 3 dpi and increased rapidly thereafter till 6 dpi indicative of active infection (Fig. 2C). The Evans Blue (EB) leakage test showed that the BBB was intact at 3 dpi, clearly indicating that the breach is not required for virus neuroinvasion. The infected mice showed barrier permeability by 6 dpi (Fig. 2D,E). Strikingly, the MTP treated mice showed a significant survival benefit (Fig. 2A,B; Appendix Fig. S3), with delayed virus invasion into the brain, and significantly lower viremia (Fig. 2C). The drug-treated mice also showed complete protection of the barrier that was comparable to control uninfected mice (Fig. 2D,E). These data demonstrate that MTP exerted a significant neuroprotective effect in the JE mouse model.

Since the BBB breach is linked to virus induced neuroinflammation, we tested the levels of several cytokines, chemokines, and interferons in the brains of infected and drug-treated mice (Fig. 2F). JEV infected mice showed very high levels of proinflammatory cytokines and chemokines: IL-6, TNF-α, RANTES, MCP-1, CXCL-1, CXCL-10, GM-CSF, IL1-β, and IFN-β starting at 3 dpi and these increased rapidly peaking at 6–7 dpi (Fig. 2F). The anti-inflammatory cytokines IL-10 and IL-12p70 along with IFN-γ were also secreted at high levels in the infected mice, indicative of active T cell infiltration in the brain during infection. Importantly, these effects were completely ameliorated in the drug-treated mice

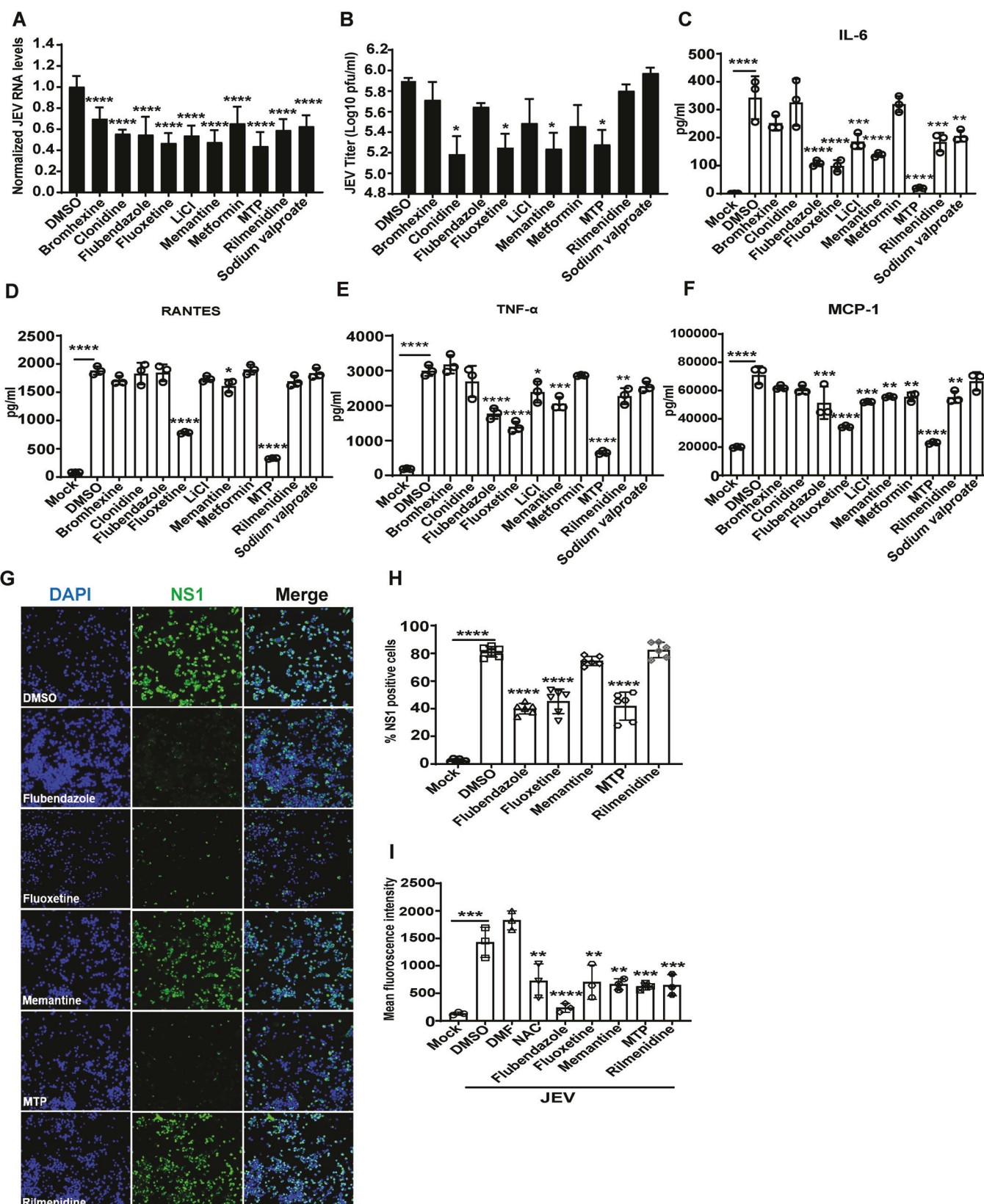

**Figure 1. Antiviral and anti-inflammatory effect of FDA-approved drugs against JEV infection.**

(A,B) Neuro2a cells were infected with JEV (MOI 1), and at 1 hpi treated with DMSO/drugs (10 μM). (A) Cells were harvested at 24 hpi and viral RNA levels were quantified using qRT-PCR. Graph shows the relative expression levels of JEV RNA normalized to DMSO-treated control. Data is plotted from two independent experiments ($n = 6$). (B) Culture supernatant was collected to determine extracellular virus titers using plaque assays. Data represents values obtained from two independent experiments ($n = 4$). (C–F) N9 cells were mock/JEV (MOI 1) infected, and at 12 hpi treated with DMSO/drugs (10 μM) for 24 h. Cytokine concentrations (pg/ml) were quantified from the culture supernatant using CBA assay. Data shows values from one representative experiment with biological triplicates ($n = 3$). (G,H) Neuro2a cells were mock/JEV (MOI 5) infected, and at 1 hpi were treated with DMSO/drugs (10 μM) till 24 hpi. (G) Cells were immunostained for JEV NS1 (green) and images were acquired on high-content imaging system. Scale bar, 10 μm. (H) Bar-graph showing percentage of NS1 positive cells from two independent experiments ($n = 6$). (I) Neuro2a cells were mock/JEV (MOI 5) infected, and at 1 hpi were treated with DMSO/FDA-drugs (10 μM); or at 16 hpi treated with DMF (70 μM)/NAC (3 mM), and maintained till 24 hpi. Post-treatment, cells were stained with oxidative stress indicator CM-H2DCFDA and fluorescence intensity was measured using flow cytometry. The graph represents mean fluorescence intensity values ($n = 3$). Data information: All data are expressed as means ± SD, statistical significance was determined using one-way ANOVA followed by Dunnett test. *$P < 0.05$; **$P < 0.01$; ***$P < 0.001$; ****$P < 0.0001$. Source data are available online for this figure.

indicative of a significant protection from neuroinflammation (Fig. 2F).

We next checked if the observed MTP-mediated anti-inflammatory response could be recapitulated in primary astrocytes which are another major source of proinflammatory cytokines in JEV-induced CNS inflammation (Fig. EV1A). At the non-toxic concentration of 10 μM (Fig. EV1B), MTP treatment resulted in a highly significant reduction of virus replication (Fig. EV1C,D), and proinflammatory cytokine/interferon secretion (Fig. EV1E). Similar trends were also observed in primary mixed glial cultures (Appendix Fig. S4A–E), and in mouse primary cortical neurons (Appendix Fig. S5A–C). Importantly, MTP could exert a similar inhibition in LPS-stimulated primary astrocytes (Fig. EV1F), mixed glial cells (Appendix Fig. S4F), primary cortical neurons (Appendix Fig. S5D), and N9 microglial cells (Appendix Fig. S6), indicating that the downregulation of proinflammatory cytokine release was not mediated entirely due to inhibition of virus replication.

Reduced neuroinvasion in drug-treated mice suggested that some protection was also conferred at the periphery. We tested different doses of MTP for toxicity on bone marrow-derived macrophages (BMDMs) (Appendix Fig. S7A). Drug treatment of JEV-infected BMDMs resulted in a significant reduction of both virus replication (Appendix Fig. S7B,C) and production of several inflammatory cytokines and interferons (Appendix Fig. S7D). A similar inhibition of proinflammatory cytokine release was also observed from LPS-treated BMDMs (Appendix Fig. S7E), indicating a strong anti-inflammatory effect of MTP.

MTP is a widely used FDA-approved antipsychotic that belongs to the phenothiazine class of drugs (Fig. 3A). MTP showed an IC50 in the range of 3–3.4 μM in mouse neuronal cells and primary cortical neurons, indicating a strong antiviral response at low doses (Fig. 3B,D). A similar reduction in virus titers was also observed (Fig. 3C,E). Encouraged by our observations, we tested another FDA-approved and widely used phenothiazine drug-Trifluoperazine (TFP) (Fig. 3A). This drug also showed robust inhibition of JEV replication with an IC50 of 2 μM (Fig. 3F), inhibition of infectious virus particles production (Fig. 3G), and a very significant block in replication complex formation (Fig. 3H,I). This drug also exerted a potent anti-inflammatory effect and significantly blocked the release of inflammatory cytokines from virus-infected microglial cells (Fig. 3J). The drug was also tested in the JE mouse model using a sublethal dose of JEV, and similar to MTP a significant survival benefit was observed (Fig. 3K). Collectively these data indicate that phenothiazines exert strong antiviral and anti-inflammatory effect for JEV infection both in vitro and in vivo.

## Phenothiazines are mTOR independent autophagy inducers

There is evidence in literature that phenothiazines are autophagy inducers (Williams et al, 2008; Zhang et al, 2007). We also assessed autophagy induction by MTP and TFP in our experimental setup, and observed that these drugs lead to rapid accumulation of lipidated MAP1LC3 (microtubule-associated protein 1 light chain 3) in Neuro2a cells (Figs. 4A,B and EV2A,B), and primary cortical neurons (Fig. 4C,D), at levels comparable to the autophagy inducer Torin1. Bafilomycin (Baf) A1 treatment in drug-treated cells led to further increase of LC3-II levels indicative of functional autophagy flux (Fig. 4E,F). This was also confirmed using GFP-LC3-RFP-ΔG expressing reporter mouse embryonic fibroblasts (MEFs) which enable high-throughput measurement of GFP/RFP ratio as an indicator of autophagy flux (Fig. EV2C,D). While high autophagy flux results in low GFP/RFP ratio as seen with Torin1 treatment, a block in flux results in a higher ratio as seen with BafA1. The GFP/RFP ratios indicated that both the drugs were inducing high autophagy flux (Fig. EV2C,D). Levels of p62 also showed a reduction similar to Torin1 both in primary cortical neurons (Fig. 4G,H), and in Neuro2a cells (Fig. 4I,J). LysoTrackerRed staining distribution and intensity in drug-treated cells was also similar to Torin1 treatment (Fig. EV2E,F), and the LysoSensor Yellow-Blue assay showed no change in lysosome acidification (Fig. EV2G,H).

A few studies have suggested that another widely used phenothiazine, chlorpromazine (CPZ) inhibits protein kinase B (PKB/Akt)/mTOR (Shin et al, 2013), and stimulates transcription factor EB (TFEB) nuclear translocation and expression of autophagy-lysosomal target genes (Zhang et al, 2017b). However, we did not observe any mTOR inactivation, as the phosphorylation of mTOR (Fig. 4I,K), and its downstream targets ribosomal protein S6 kinase beta-1 (p70S6K) (Fig. 4L,M) and eukaryotic translation initiation factor 4E-binding protein 1 (4EBP1) (Fig. 4N,O), remained unaffected. These drugs also did not lead to any significant TFEB nuclear translocation as was observed with Torin1 treatment (Fig. 4P,Q). These data suggested that MTP and TFP induce autophagy through an mTOR-independent mechanism.

## Methotrimeprazine induces adaptive ER stress and dysregulates ER calcium signaling

We next checked if MTP induced any changes in the transcript levels of autophagy genes, and observed enhanced levels of Atg12, Atg16L1, LC3A, and LC3B in neuronal cells (Fig. 5A). Since JEV

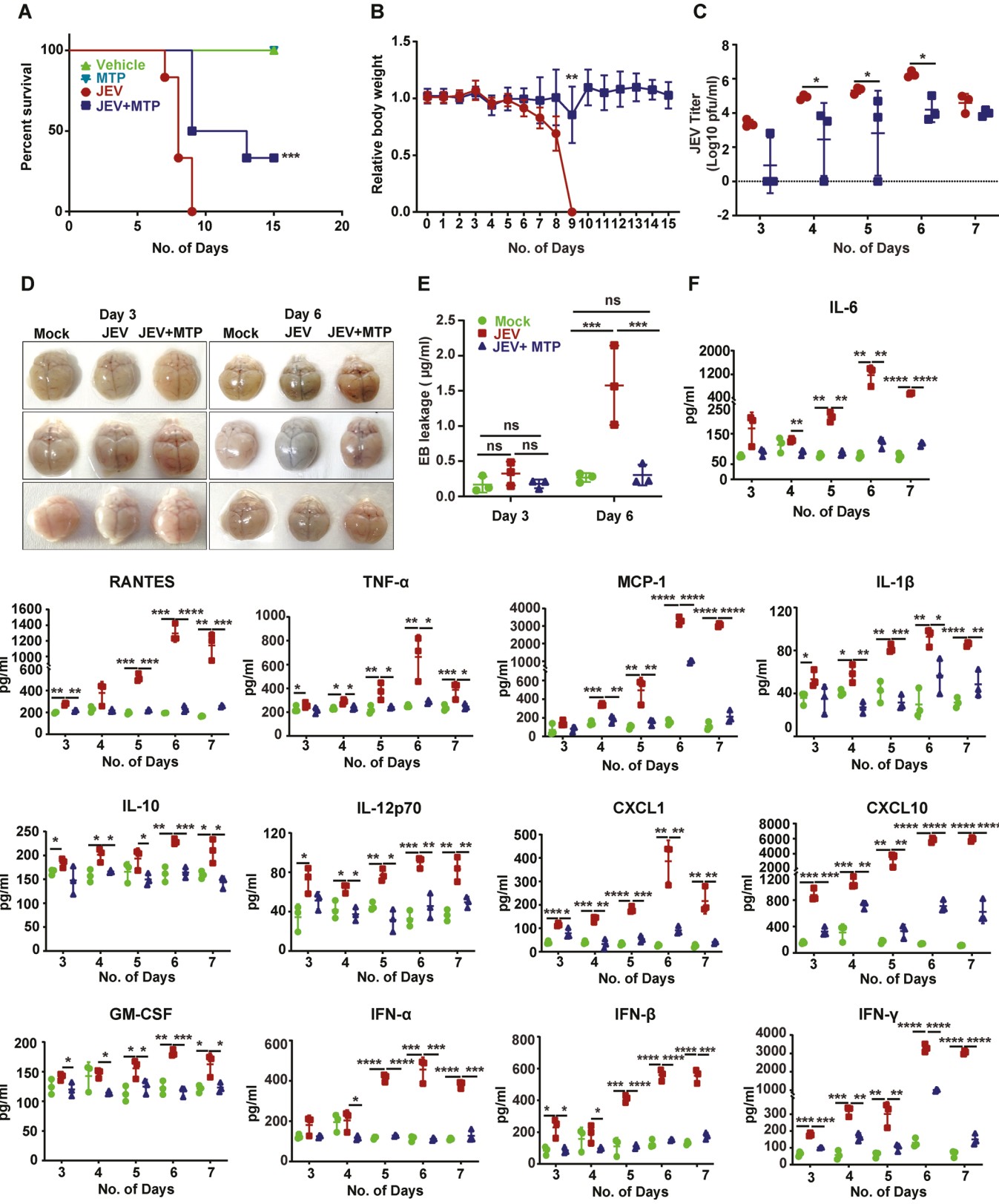

**Figure 2. Efficacy of MTP in JEV-mouse model.**

Three-week-old C57BL/6 mice were mock/JEV-S3 ($10^7$ pfu) infected through an i.p. injection, and at 4 hpi were treated with vehicle control (PEG400) or MTP (2 mg/kg) by oral gavage at an interval of 24 h for 15 days. All mice were monitored for the appearance of encephalitis symptoms until death. (A,B) (A) Survival curve of mock ($n = 4$)/MTP ($n = 4$)/JEV ($n = 6$)/JEV + MTP ($n = 6$), Log-rank (Mantel-Cox) test was used to determine the statistical significance comparing JEV and JEV + MTP mice group. (B) Graph representing the change in body weight of vehicle/MTP-treated infected mice group normalized to mock-infected mice group, compared by unpaired Student t-test. (C) Mock or Vehicle/MTP treated infected mice ($n = 3$/time point from each group) were sacrificed at indicated time points. Brain tissues were homogenized and the supernatant was used for JEV titration using plaque assay. Each data point denotes one mouse, and the virus titers between JEV and JEV + MTP group was compared by unpaired Student t-test. (D,E) Mock or Vehicle/MTP treated infected mice received an i.p. injection of 2% Evans blue dye (100 µl). Mice were sacrificed at 3 and 6 dpi. (D) Representative images showing Evans blue dye distribution in the brain ($n = 3$). (E) The brain tissues were homogenized in DMF (200 mg/500 µl DMF) and absorbance was measured at 620 nm. The concentration of Evans blue was quantitated according to standard curve and significance was compared by two-way ANOVA followed by Tukey test. (F) Brain tissue was collected at indicated time points, and an equal amount of protein (30 µg) from each sample was used for the quantitation of cytokine levels using CBA. Data were analyzed with LEGENDplex™ Multiplex assay software, and significance was compared by one-way ANOVA followed by Dunnett test ($n = 3$/time point from each group). Data information: All data expressed as means ± SD. *$P < 0.05$; **$P < 0.01$; ***$P < 0.001$; ****$P < 0.0001$. Source data are available online for this figure.

infects and replicates efficiently in fibroblasts and epithelial cells in the periphery, we also examined MTP-induced changes in MEFs. These showed a similar transcriptional upregulation of autophagy genes, along with Bcl2 (Fig. EV3A). Interestingly, both cell types showed downregulation of Atg13, Atg14, Atg3, Atg4a, Atg4b, and Atg9 (Figs. 5A and EV3A). Transcriptional upregulation of Atg12 and LC3 has been reported to specifically occur through activated protein kinase R-like ER kinase (PERK)/activating transcription factor 3 (ATF4), hinting that autophagy activation by MTP could be a result of ER stress activation (B'Chir et al, 2013; Kroemer et al, 2010).

We tested other parameters of the UPR in drug-treated cells. A primary response is the PERK-mediated phosphorylation of eIF2α, that inhibits ribosome ternary complex recycling and attenuates protein translation. Thapsigargin-treated Neuro2a cells showed very rapid and robust eIF2α phosphorylation starting at 1 h of treatment, which was sustained at 3 h, but declined thereafter and returned to baseline by 12 h (Fig. 5B,C). These cells also showed PERK phosphorylation (indicated by mobility shift of PERK) till 6 h (Fig. 5B). PERK activation also results in enhanced ATF4 levels that activate a transcriptional response program that can either be adaptive (through autophagy induction) or apoptotic (through production of C/EBP homologus protein: CHOP and activation of pro-apoptotic proteins) (Matsumoto et al, 2013; McCullough et al, 2001). The other two ER stress sensors are IRE1α that activates X-box binding protein 1 (XBP1) through an unconventional splicing (XBP1 spl), and activating transcription factor 6 (ATF6) that is cleaved in the Golgi to generate the ATF6 (N) transcriptional factor. In Thapsigargin-treated Neuro2a cells, CHOP protein levels were detectable by 6–12 h (Fig. 5B,C), and significant transcriptional activation of ATF4, CHOP, glucose-regulated protein 78 (GRP78), XBP1, XBP1(s), and ATF6 was observed (Fig. 5D).

MTP-treated Neuro2a cells showed a modest enhancement of eIF2α phosphorylation (Fig. 5B,C). These cells also showed activation of ATF4, CHOP, XBP1(s), and ATF6, but no activation of DNAJC, GRP78, and XBP1 at 6 h of treatment and down-regulation of these transcripts by 12 h (Fig. 5D). No CHOP protein expression was observed on MTP treatment (Fig. 5B,C). The modest transcriptional activation of CHOP and absence of its protein expression on MTP treatment suggested the activation of an adaptive but not pro-apoptotic pathway of UPR induction.

Thapsigargin-treated MEFs showed enhanced ER stress responses similar to those observed in Neuro2a cells (Fig. EV3B–D). On the other hand, MTP-treated MEFs showed marginal

eIF2α phosphorylation, no detectable CHOP protein, and comparatively low transcriptional activation of the other genes (Fig. EV3D; Appendix Fig. S8A). A similar transcriptional profile was also observed in MTP-treated primary astrocytes (Appendix Fig. S9A).

We also examined the ER stress signatures in virus-infected and drug-treated cells. In agreement with our earlier study on neuronal cells (Sharma et al, 2017), JEV-infected MEFs showed significant eIF2α phosphorylation (Fig. EV3B,C), and upregulation of CHOP, GRP78, DNAJC3, ATF4, XBP1, and XBP1 spl. at 12 h of infection (Appendix Fig. S8A). However, ER stress transcripts were high at early time points only in drug-treated and infected cells (Appendix Figs. S8A and S9A), suggesting that the drug induces the activation of an early adaptive ER stress response preceding virus replication. MTP treatment also significantly reduced transcripts of inflammatory cytokines, type I IFN, and ISGs in both mock and virus-infected cells (Appendix Fig. S8B,D). Interestingly, an upregulation of cholesterol metabolic pathway genes was observed (Appendix Fig. S8C). Collectively, these data show that MTP treatment establishes a state of adaptive ER stress in cells at early time points.

One of the most critical contributors to ER stress induction is dysregulated ER $Ca^{2+}$ homeostasis. It is important to note that perturbations in ER $Ca^{2+}$ signaling can lead to a variety of viral pathogenesis and therefore, it is emerging as a potential therapeutic target (Saurav et al, 2021; Sultan et al, 2022). Since MTP induces ER stress response, we examined if it can modulate ER $Ca^{2+}$ signaling. We performed live cell $Ca^{2+}$ imaging using ratiometric FURA-2AM dye in Neuro2a, MEFs, and primary astrocytes. The fluorescence intensity of FURA-2AM corresponds to cytosolic $Ca^{2+}$ levels. We first performed a dose–response assay in MEFs with increasing concentration of MTP in absence of extracellular $Ca^{2+}$. We observed that 10 µM MTP can induce an increase in cytosolic $Ca^{2+}$ levels and 100 µM MTP completely depletes Thapsigargin (Tg) sensitive $Ca^{2+}$ stores (Fig. EV3E). As these imaging assays were performed without $Ca^{2+}$ in the extracellular bath, it suggests that the source of this rise in cytosolic $Ca^{2+}$ levels is intracellular stores. ER is the major source of intracellular $Ca^{2+}$ stores. Therefore, we examined if MTP is driving ER $Ca^{2+}$ release to cytosol. We repeated the live cell $Ca^{2+}$ imaging experiments with 10 µM MTP in absence of extracellular $Ca^{2+}$ and observed a rise in cytosolic $Ca^{2+}$ levels in Neuro2a cells (Fig. 5E), and MEFs (Fig. EV3F). In the same experiments, we then added Tg to block SERCA channels present on the ER, which led to further increase in cytosolic $Ca^{2+}$ levels

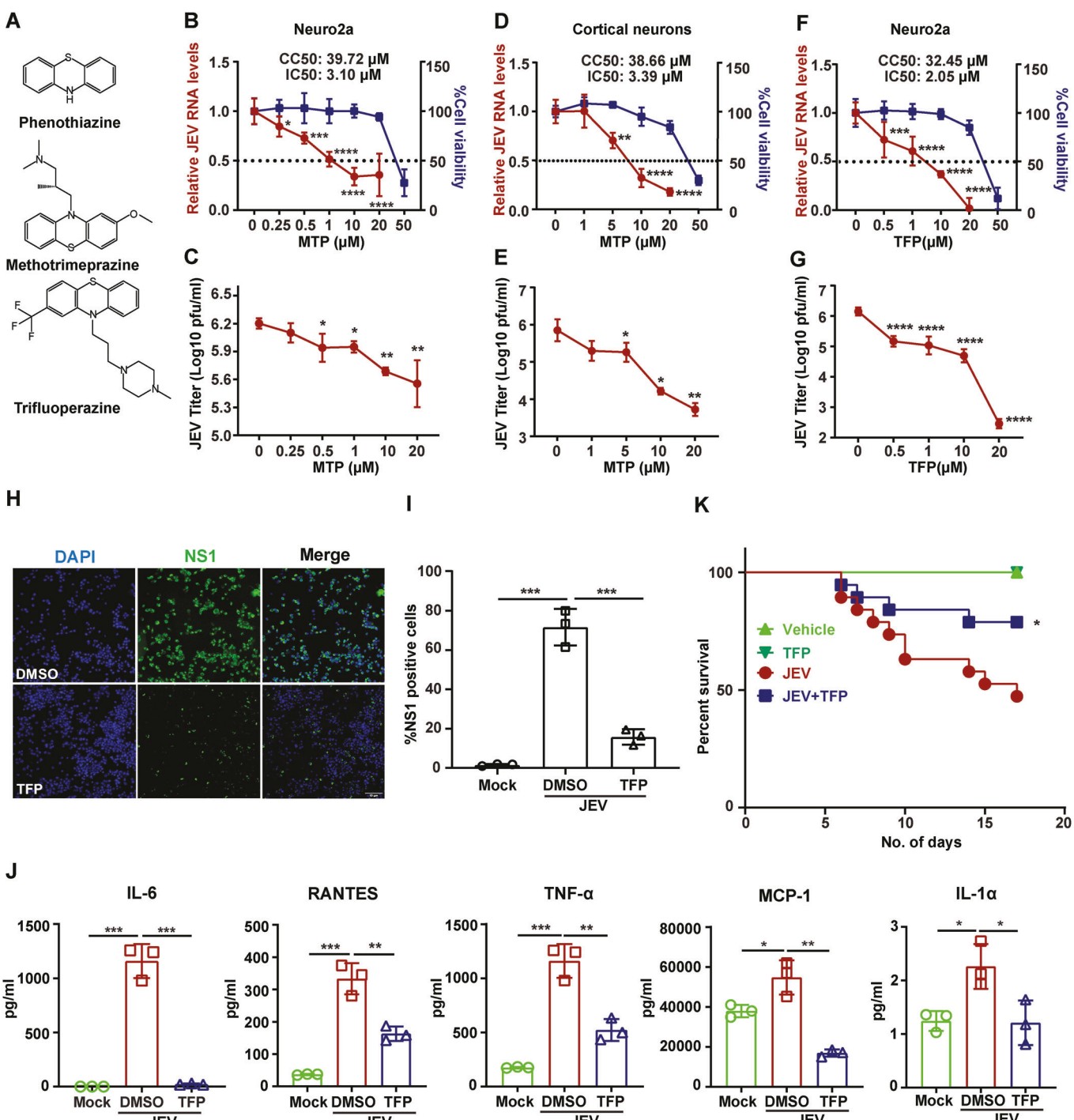

(Figs. 5E and EV3F). This suggests that MTP can only partially mobilize ER Ca$^{2+}$ stores. We next performed opposite experiments wherein we first stimulated ER Ca$^{2+}$ release with Tg and then gave MTP treatment (Figs. 5F and EV3G). If MTP induces Ca$^{2+}$ movement from ER, then amplitude of this Ca$^{2+}$ mobilization should be substantially decreased after Tg stimulation. Indeed, we observed that post Tg stimulation, MTP-mediated cytosolic Ca$^{2+}$ rise was significantly decreased (Figs. 5F,G and EV3G,H). Likewise, Tg induced ER Ca$^{2+}$ release was drastically reduced upon pre-

stimulation with MTP (Figs. 5E,H and EV3F,I). We performed similar experiments in the primary astrocytes and observed the same phenomenon in them as well (Appendix Fig. S9B–E). Taken together, these experiments demonstrate that MTP and Tg mobilize Ca$^{2+}$ from same intracellular stores i.e., ER; most likely by acting over same target viz. SERCA channels. Further, these experiments establish MTP as a potent ER Ca$^{2+}$ release inducer and that in turn at least partially explain the molecular mechanism that connects MTP treatment to induction of ER stress.

◄

**Figure 3. Phenothiazines exert antiviral and anti-inflammatory effects.**

(A) Chemical structure of the phenothiazine ring and its derivatives Methotrimeprazine (MTP) and Trifluoperazine (TFP). (B–E) Neuro2a cells (B,C), and cortical neurons (D,E), were mock/JEV (MOI 1) infected, and at 1 hpi were treated with DMSO/MTP at indicated concentrations till 24 hpi. Cells were harvested, qRT-PCR was performed for the quantitation of JEV RNA levels, and % cell viability was measured using MTT assay. Data were normalized to DMSO control (n ≥ 3), and compared between DMSO and MTP treated JEV-infected cells. Graphs represent the CC50 and IC50 values (B,D). Virus titers in the culture supernatants of Neuro2a cells (C), cortical neurons (E) was determined by plaque assays (n = 3). Statistical significance was determined using one-way ANOVA followed by Dunnett test. (F,G) Neuro2a cells were mock/JEV (MOI 1) infected and treated with DMSO/TFP at the indicated concentrations till 24 hpi. CC50 and IC50 were calculated as described above. (G) Culture supernatant was collected and virus titers was determined using plaque assay (n = 3). Statistical significance was determined using one-way ANOVA followed by Dunnett test. (H,I) Neuro2a cells were mock/JEV (MOI 5) infected, and at 1 hpi treated with 10 μM TFP till 24 hpi. Cells were immunostained with JEV NS1 antibody, and images were visualized by high-content imaging system, representative images are shown. Scale bar, 10 μm (H). (I) Graph shows % NS1 positive cells (n = 3), unpaired Student t-test. (J) N9 cells were infected with JEV (MOI 1), at 12 hpi cells were treated with TFP (10 μM) for 24 h. Cytokines were quantified from the soup using CBA (n = 3), unpaired Student t-test. (K) C57BL/6 (3 weeks old) mice were infected by i.p. injection of DMEM (mock) or JEV (10⁶ pfu), and at 4 hpi treated with vehicle control (PEG400)/TFP (1 mg/kg) by oral route at an interval of 24 h till 15 days. All mice were monitored for JEV symptoms until death. The survival curve of mock (n = 4)/TFP (n = 4)/JEV (n = 19)/JEV + TFP (n = 19) was plotted from two independent experiments. Log-rank (Mantel-Cox) test was used to determine the statistical significance comparing vehicle and drug-treated infected mice group. Data information: All data expressed as means ± SD. *P < 0.05; **P < 0.01; ***P < 0.001; ****P < 0.0001. Source data are available online for this figure.

## MTP induced ER stress is crucial for autophagy and antiviral effect

We next checked if pharmacological rescue of ER stress using the chemical chaperone 4-PBA (de Almeida et al, 2007), would impact MTP-mediated autophagy induction and antiviral effect (Fig. EV4). In Neuro2a cells, 4-PBA treatment along with MTP (Fig. EV4A), significantly reversed LC3-II lipidation (Fig. EV4B,C), indicating that MTP-mediated autophagy induction is mediated through activation of ER-stress response. 4-PBA also completely rescued JEV replication as seen by enhanced JEV RNA levels (Fig. EV4D), and titers (Fig. EV4E), indicating that ER-stress is critical for the observed antiviral effect of MTP.

## MTP targets JEV replication

We next attempted to elucidate which step of the virus life-cycle was being targeted by phenothiazines (Fig. 6A). MTP did not lead to any reduction in virus entry (1 hpi) as measured through JEV RNA levels in primary cortical neurons (Fig. 6B), and Neuro2a cells (Fig. 6C). A quantitative immunofluorescence assay of JEV envelope antibody labeled virus particles in control and MTP treated EGFP-LC3 Neuro2a cells was performed (Fig. 6D). As expected, the number of GFP-LC3 puncta were significantly increased in drug-treated cells (Fig. 6D,E), however, no overlap of labeled virus particles with any autophagosome was observed (Fig. 6D), and the number of endocytosed virions per cell showed no difference between the DMSO/MTP treated condition (Fig. 6D,F). These data suggested that the drug treatment does not inhibit virus entry, and the endocytosed virus particles do not appear to be targeted for virophagy.

We next performed a time course analysis of virus infection in control and drug-treated Neuro2a cells (Fig. 6G–I,L,M), primary cortical neurons (Fig. 6J–K), MEFs (Fig. EV5A,B), and HeLa cells (Appendix Fig. S10A,B). JEV life-cycle begins with a low endosomal pH mediated uncoating of the virus envelope, followed by nucleocapsid release into the cytosol, capsid dissociation, and translation of the plus-strand viral RNA into a single polyprotein on the ER, that subsequently gives rise to virus structural and nonstructural proteins. Depending on the MOI and cell type this process takes 3–6 h, during which time the viral RNA levels decrease compared to 1 hpi, likely due to degradation of a fraction of the endocytosed virus in the endosomal system (Figs. 6G,L and

EV5A; Appendix Fig. S10A). The viral RNA levels were identical between DMSO vs MTP treated cells till 3–6 hpi, suggesting that MTP does not exert an antiviral effect at early time points of the virus infection process. Once the incoming viral RNA is translated, the virus replication complex is established on ER-derived membranes, where dsRNA replicative intermediates are generated, and a rapid increase in production of virus plus-strand RNA is seen (Nain et al, 2017; Sehrawat et al, 2021). A fraction of this plus-RNA is diverted for translation to generate more structural and nonstructural proteins, which increases the number of virus replication complexes in the cell. Another pool of the plus-RNA genome is packaged into virions that undergo maturation in the TGN for egress as infectious virus particles (Nain et al, 2017; Sehrawat et al, 2021). A comparison of virus replication kinetics between control and drug-treated cells clearly showed that the antiviral effect was exerted either at the level of replication complex formation which was severely compromised in drug-treated neuronal cells (Fig. 6G–K), MEFs (Fig. EV5A,B), and HeLa cells (Appendix Fig. S10A,B). A significant reduction in JEV negative-strand RNA was observed (Fig. 6H), clearly indicating reduced formation of virus ds RNA containing replication complexes in MTP-treated cells.

## Phenothiazines exert an autophagy-dependent antiviral effect, but do not impact viral RNA translation

To examine if the observed antiviral effect of these drugs was autophagy-dependent we generated ATG5 and ATG7 (knockout) Neuro2a cells through CRISPR-Cas9 system (Fig. 7A,D). While WT-infected cells treated with MTP showed a dose-dependent reduction in JEV RNA levels and titers, the autophagy-deficient cells showed a complete block of this antiviral effect (Fig. 7B,C,E,F). Similar results were seen in ATG5 KO MEFs (Fig. EV5C–E; Appendix Fig. S11A–C), ATG5 KO HeLa cells (Appendix Fig. S10C–E), and in Atg7 depleted MEFs (Appendix Fig. S11D–F). These data clearly indicated that the antiviral effect of the phenothiazines in diverse cell lines was mediated through autophagy.

We next attempted to address how phenothiazine induced autophagy was exerting an antiviral effect. Our earlier studies have shown that non-lipidated LC3 (LC3-I) is a crucial host factor for the biogenesis of the JEV replication complex, and for virus replication (Sarkar et al, 2021; Sharma et al, 2014). We have also

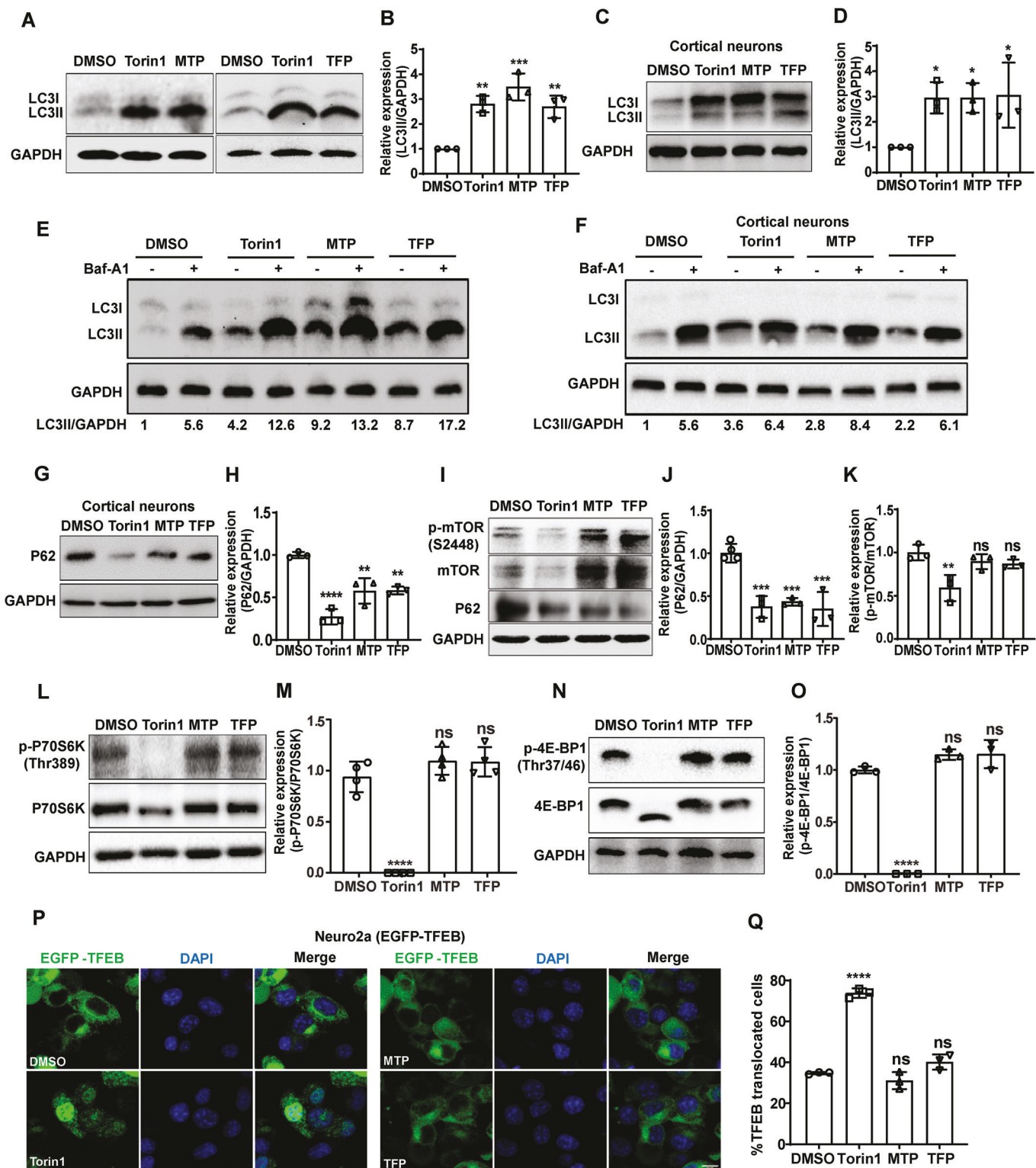

found no evidence for the degradation of the virus nonstructural proteins, or the replication complex through canonical autophagy (Sharma et al, 2014).

To test if phenothiazine induced autophagy was impacting translation of viral RNA, we performed polysome profiling

(Figs. 7G–L and EV5F–K; Appendix Fig. S12). Clear 40S, 60S subunits, 80S monosome, and several polysome peaks were observed in all the profiles. Thapsigargin treatment known to result in global protein synthesis inhibition, significantly altered the polysome profile, with decreased P/M ratios in both WT and ATG7

**Figure 4. Phenothiazines are mTOR-independent autophagy inducers.**

(A–D) Neuro2a (A,B), and cortical neurons (C,D), were treated with DMSO/Torin1 (1 µM)/MTP (10 µM)/TFP (10 µM) for 6 h. Protein lysates were immunoblotted for LC3 and GAPDH (loading control). (B,D) Bar-graph shows relative expression of LC3II/GAPDH normalized to DMSO control from three independent experiments. (E,F) Neuro2a (E), and cortical neurons (F), were treated with DMSO/Torin1 (1 µM)/MTP (10 µM)/TFP (10 µM) for 4 h, followed by Baf-A1 (100 nM) treatment for 2 h. The values below the blot show relative levels of LC3II/GAPDH protein after normalization to DMSO-treated cells. (G,H) Cortical neurons were treated as described in panel (A), protein lysates were immunoblotted for P62 and GAPDH (loading control). (H) Bar-graph shows relative expression of P62/GAPDH normalized to DMSO control from three independent experiments. (I–O) Neuro2a cells were treated with DMSO/Torin1 (1 µM)/MTP (10 µM)/TFP (10 µM) for 6 h, protein lysates were analyzed by western blotting with P62 (I,J), p-mTOR (S2448), mTOR (I–K), p-P70S6K (Thr386), p70S6K (L,M), p-4E-BP (Thr37/46), 4E-BP (N,O), and GAPDH (loading control) antibodies. (J,K,M,O) Bar-graphs show relative protein expression level calculated after normalization to DMSO control from three or more independent experiments. (P,Q) EGFP-TFEB expressing Neuro2a cells were treated with DMSO/Torin1 (1 µM)/MTP (10 µM)/TFP (10 µM) for 6 h. Images were acquired on high content imaging system and representative images are shown. Scale bar, 10 µm. (Q) Bar-graph showing the percentage EGFP-TFEB nuclear translocation ($n = 3$). Data information: All data were expressed as means ± SD, one-way ANOVA followed by Dunnett test was used to calculate statistical significance *$P < 0.05$; **$P < 0.01$; ***$P < 0.001$; ****$P < 0.0001$; ns, non-significant. Source data are available online for this figure.

KO Neuro2a cells (Appendix Fig. S12A,B). MTP treatment did not alter the P/M ratios, and thus does not appear to majorly impact global protein translation. We next checked the polysome profiles of mock, JEV, JEV/MTP treated WT, and ATG7 KO Neuro2a cells at 6 hpi (Fig. 7G,J). No major changes were seen in the P/M ratios between all three conditions. The percentage distribution of JEV and GAPDH mRNA across the gradient was analyzed. GAPDH mRNA was most abundant in the polysome fractions in all conditions (Fig. 7H,K; Appendix Table S2). While ~20–28% of JEV RNA was associated with the 80 S monosome fraction, a major part was present in the actively translating fractions in both control and drug-treated conditions, in WT and ATG7 KO Neuro2a cells (Fig. 7I,L; Appendix Table S2). Similar profile was also observed in WT and ATG5 KO MEFs (Fig. EV5F–K; Appendix Table S2). These data are consistent with the notion that translation of JEV RNA is not majorly altered by MTP treatment, or in autophagy-deficient conditions, and point towards a direct role for autophagy modulated host-factors in the generation and/or maintenance of the virus replication complex.

## MTP induced autophagy reduces proinflammatory cytokine release and NLRP3 levels in microglial cells

Microglial cells are crucial to modulate neuroinflammation through secretion of proinflammatory cytokines. We next checked if this process was autophagy-dependent in the context of JEV infection (Appendix Fig. S13). Secretion of proinflammatory cytokines IL-6, TNF-α, RANTES, and MCP-1 from JEV infected microglial cells increased with increasing MOI (Appendix Fig. S13B). Depletion of Atg5 in these cells resulted in significantly higher levels of cytokine secretion, indicating a crucial role of autophagy in mediating neuroinflammation (Fig. 7A,B; Appendix Fig. S13A,B). Autophagy-deficient microglial cells also displayed significantly enhanced JEV induced cell death (Appendix Fig. S13C), however, the JEV titers between autophagy competent and deficient microglial cells were not significantly different (Appendix Fig. S13D).

We then checked if the anti-inflammatory effect exerted by MTP in microglial cells was autophagy-dependent (Fig. 8A). As shown earlier (Fig. 1C–F; Appendix Fig. S6), MTP treatment resulted in significantly reduced levels of cytokine release from JEV infected cells (Fig. 8B). MTP treatment of Atg5 depleted N9 cells resulted in reduced cytokine release however these levels were still significantly higher compared to MTP treated wild-type cells. JEV infection resulted in increased protein levels of NLRP3 in both NT/Atg5 siRNA-treated microglial cells, indicating activation of the

inflammasome (Fig. 8C,D). In infected/LPS-stimulated cells, MTP treatment significantly reduced NLRP3 protein levels in siNT, but not in Atg5 depleted condition (Fig. 8C–F). This indicated autophagy-dependent reduction of NLRP3, which could contribute reduced inflammasome activation and decreased cytokine secretion.

## Discussion

One-third of JE infections are fatal, and one-third develop permanent cognitive and/or motor defects due to severe neurological damage. The disease is acute and its pathogenesis is a combination of direct virus induced neuronal cell death and a massive neuroinflammatory response. Suppression of neuroinflammation in the patient is likely to be critical to improving prognosis, and this necessitates the need for development of effective therapies.

Our studies on JEV-host interactions have shown that infection-induced ER stress and autophagy are closely linked to virus replication and neuronal death (Sharma et al, 2014; Sharma et al, 2017). Several FDA approved drugs have been shown to enhance autophagy, and this provides a strong rationale for repurposing these drugs for treatment of diseases where autophagy upregulation could potentially provide a therapeutic benefit.

Based on published literature we curated a pool of FDA-approved drugs with autophagy-inducing potential and tested these for both antiviral and anti-inflammatory effects in vitro. Further, we investigated promising in vitro leads in the JE mouse model, at oral doses equivalent to the recommended human dose. The antipsychotic phenothiazine drug MTP showed significantly improved survival, reduced neuroinvasion and complete protection against BBB breach and neuroinflammation. MTP also known as Levomepromazine (brand name Nozinan) is prescribed for relief of moderate to severe pain and anxiety. Another widely prescribed antipsychotic phenothiazine, TFP showed similar antiviral and neuroprotective effects in vitro and in vivo. Our studies suggest that the antiviral and neuroprotective mechanism is likely to be a complex interplay of drug-induced dysregulation of ER Ca²⁺ homeostasis, adaptive ER stress, and autophagy. Given the broad role of autophagy in diverse cellular processes, indeed, it is challenging to establish a single drug target, and ascertain a linear relationship in the sequence of events.

Drugs of the phenothiazine family are typical antipsychotics that are widely used in clinical practice for the treatment of bipolar

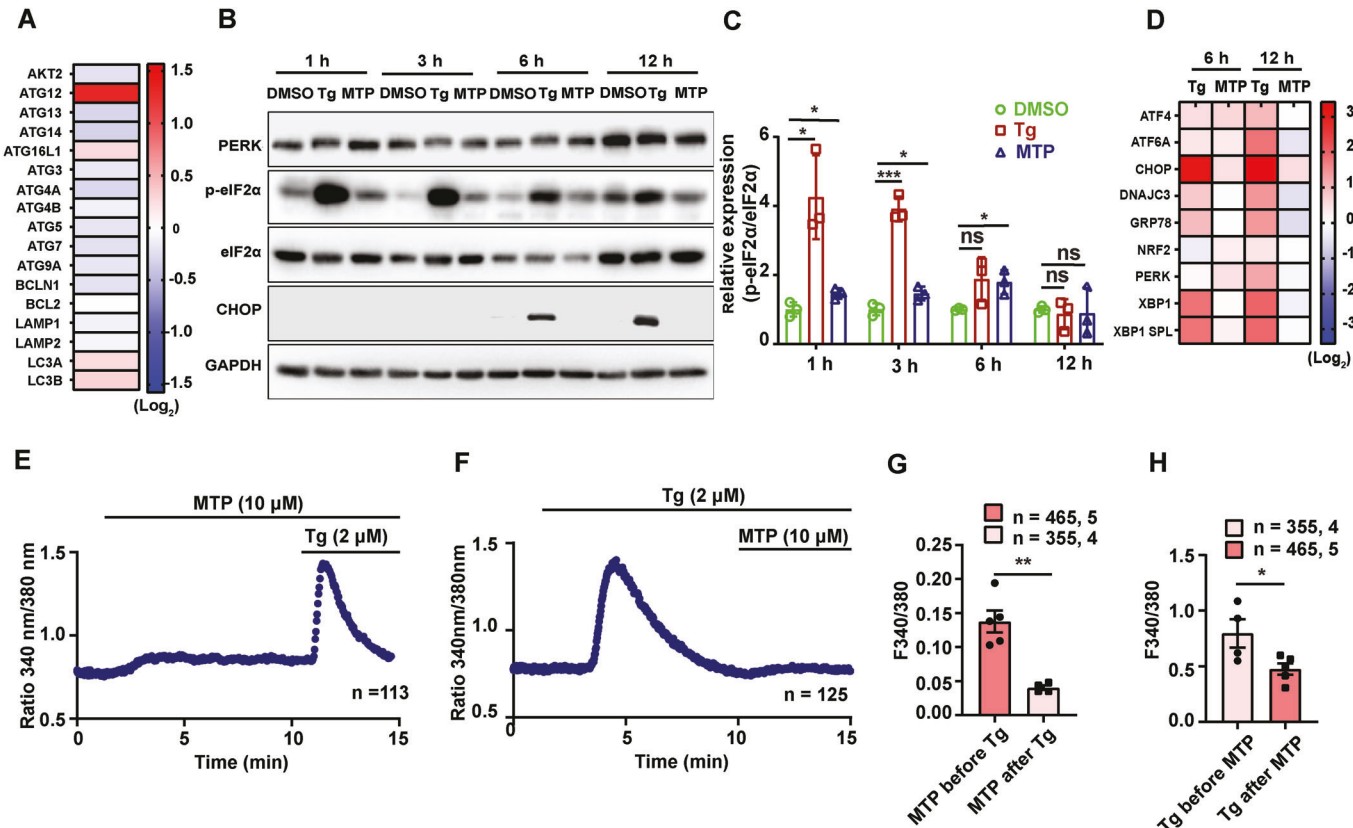

**Figure 5. MTP activates adaptive ER stress and dysregulates ER calcium homeostasis.**

(A) Neuro2a cells were treated with MTP (10 μM) for 6 h and mRNA levels of autophagy genes were determined by qRT-PCR. Heatmap shows relative gene expression level after normalization to DMSO-treated control ($n = 6$). (B,C) Neuro2a cells were treated with DMSO/Tg (1 μM)/MTP (10 μM) for the indicated time points. (B) Protein lysates were immunoblotted for PERK, p-eIF2α, eIF2α, CHOP, and GAPDH (loading control). (C) Bar-graph shows relative expression of p-eIF2α/eIF2α normalized to DMSO control from three independent experiments. (D) mRNA levels of ER stress markers and chaperones were quantified using qRT-PCR. Heatmap depicts relative gene expression normalized to DMSO control, represented as mean ($n = 3$). (E) Representative Ca$^{2+}$ imaging trace of experiments where Neuro2a cells were stimulated with 10 μM MTP in absence of extracellular Ca$^{2+}$ followed by addition of 2 μM thapsigargin (Tg). Here, "$n = 113$" denotes the number of cells in that particular trace. (F) Representative Ca$^{2+}$ imaging trace of experiments where cells were stimulated first with 2 μM Tg to deplete ER Ca$^{2+}$ stores, followed by addition of 10 μM MTP in absence of extracellular Ca$^{2+}$. Here, "$n = 125$" denotes the number of cells in that particular trace. (G) Quantitation of MTP (10 μM) induced ER Ca$^{2+}$ stores depletion before and after the addition of 2 μM Tg. 465 and 355 cells from 5 and 4 independent imaging dishes were analyzed for the two conditions, respectively. (H) Quantitation of Tg (2 μM), induced ER Ca$^{2+}$ stores depletion before and after the addition of 10 μM MTP. 355 and 465 cells from 4 and 5 independent imaging dishes were analyzed for the two conditions, respectively ("$n = x, y$" where "$x$" denotes total number of cells imaged and "$y$" denotes number of traces recorded). Data information: In (G,H), data presented are means ± S.E.M., Unpaired Student t-test. $*P < 0.05$; $**P < 0.01$; $***P < 0.001$; $****P < 0.0001$; ns, non-significant. Source data are available online for this figure.

disorders, psychosis, and schizophrenia. The antipsychotic effect is attributed to the blockage of dopamine D2 receptors in the brain. The neuroprotective effects of these drugs in vitro and rodent models of Alzheimer's, Parkinson's, and Huntington's disease are well documented (Hasegawa, 2006; Makhaeva et al, 2019; Sontag et al, 2012; Tucker et al, 2018; Varga et al, 2017; Yang et al, 2017), and has also been linked to autophagy induction (Congdon et al, 2012). Phenothiazine hydrochloride was first identified as an autophagy inducer in a high-throughput drug screening assay using a *C. elegans* model of protein aggregation (Gosai et al, 2010). Stress-dependent pharmacological activation of autophagy through TFP has been shown to have neuroprotective effects under conditions of α-synuclein accumulation in human dopaminergic neurons (Hollerhage et al, 2014), and in Pink1 deficient zebrafish model and human cells (Zhang et al, 2017b).

Phenothiazines show diverse biological effects ranging from anti-cancer to anti-pathogen (virus, bacteria, fungus, protozoa)

(Choudhary et al, 2018; Golden et al, 2021; Otreba et al, 2020; Walsh et al, 2020). CPZ has shown antiviral effects against several viruses, including SARS-CoV-2 and flaviviruses such as JEV, DENV, and WNV. Prochlorperazine has also shown antiviral activities against JEV, DENV, HCV, and EBOV (Madrid et al, 2015; Otreba et al, 2020; Simanjuntak et al, 2015). These drugs have been shown to alter cellular lipid dynamics (Chamoun-Emanuelli et al, 2013), or obstruct endocytic pathways (Gao et al, 2019; Hashizume et al, 2023; Nawa et al, 2003; Nemerow and Cooper, 1984). Besides acting as host-directed antivirals, phenothiazines have also been shown to directly interact with and destabilize the EBOV glycoprotein (Zhao et al, 2018).

The autophagy inducing properties of phenothiazines are documented though the mechanistic details lack clarity (Shin et al, 2013; Williams et al, 2008; Zhang et al, 2007). Studies in different cancer lines have shown both autophagy induction (Chu et al, 2019; Jhou et al, 2021; Matteoni et al, 2021; Qian et al, 2019;

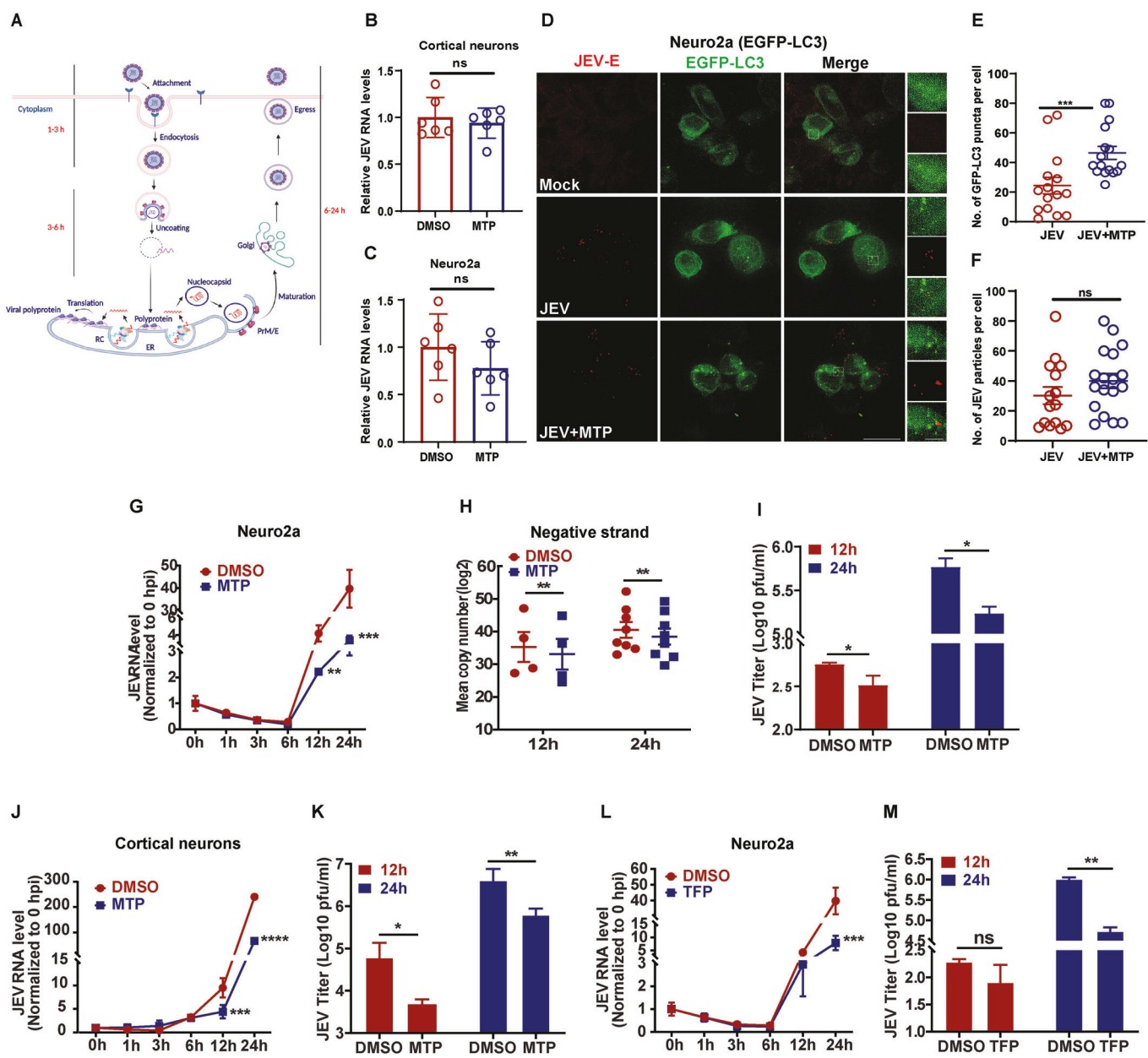

Shin et al, 2013; Wu et al, 2016), and autophagy flux inhibition (Johannessen et al, 2019; Li et al, 2020; Xia et al, 2021; Zhang et al, 2017a). CPZ is known to inhibit PI3K/Akt/mTOR in glioma and oral cancer cells (Jhou et al, 2021; Shin et al, 2013), and also induce ER stress in glioblastoma cell lines (Matteoni et al, 2021). However, most studies showing phenothiazine induced mTOR inhibition and cancer cell cytotoxicity have used high drug doses in the range of 20-50 μM and beyond (Jhou et al, 2021; Medeiros et al, 2020; Shin et al, 2013). A structure-function relationship study identified a pharmacore that could induce neuronal autophagy in an Akt-and mTOR-independent manner. This was defined as the $N^{10}$-substituted phenoxazine/phenothiazine, whereas the non-substituted phenoxazine and phenothiazine did not stimulate autophagy (Tsvetkov et al, 2010). In our study, MTP and TFP were potent autophagy inducers in several cell types and resulted in

the activation of functional autophagy flux. Autophagy induction was observed to be mTOR independent and there was no TFEB translocation into the nucleus. MTP treatment also resulted in the transcriptional activation of genes involved in autophagosome expansion and formation (Atg12, Atg16l1, LC3A, LC3B).

Our results demonstrate that phenothiazine induces adaptive ER stress and autophagy. One of the most critical signaling events that regulates both ER stress and autophagy is increase in intracellular $Ca^{2+}$ levels (Hoyer-Hansen et al, 2007; Lim et al, 2023). Therefore, we examined the role of MTP on cellular $Ca^{2+}$ signaling and found that MTP treatment results in rise in cytosolic $Ca^{2+}$ levels in diverse cell types. Our detailed live cell $Ca^{2+}$ imaging experiments showed that the source of this increase in cytosolic $Ca^{2+}$ is ER $Ca^{2+}$ release. Since Tg (SERCA inhibitor) and MTP treatment showed a non-additive rise in $Ca^{2+}$ levels, it suggests that they are mobilizing $Ca^{2+}$

◀ **Figure 6. MTP targets JEV replication.**

(A) Schematic depicting time course of the JEV infection process (Created with BioRender.com). Viral genome translation initiates between 3–6 h, and with the help of viral nonstructural and host proteins the replication complex (RC) is generated on ER membranes. The dsRNA replicative intermediate is synthesized here, through which several copies of the positive-sense JEV RNA are made. These are used for (i) translation to generate more viral proteins and RCs, (ii) packaging into virus particles that undergo maturation and egress. (B,C) Cortical neurons (B), Neuro2a cells (C), were pre-treated with DMSO/MTP (10 μM) for 1 h, and then infected with JEV (MOI 5) for 1 h in the presence of the drug. Cells were given trypsin treatment to remove dish/cell bound virus particles, and the levels of internalized virus were measured using qRT-PCR. Data is plotted from two independent experiments ($n = 6$), means ± SD, unpaired Student t-test. (D,F) Neuro2a cells stably expressing EGFP-LC3 were grown on glass coverslips and were mock/JEV (MOI 50) infected. At 1 hpi, cells were given DMSO/MTP treatment for 1 h. Cells were immunostained with JEV envelope antibody, and SIM imaging was peformed. The right panel shows magnified view of the region marked by rectangle. Scale bar, 10 μm, 5 μm (inset). Graph shows quantitation of autophagosome number (green puncta) per cell (E), and JEV particles (red dots) per cell (F), calculated from 15 to 18 cells across two independent coverslips. Quantification was performed using Imaris 8 software and expressed as means ± SEM. Statistical analysis was performed using unpaired Student t-test. (G–I) Neuro2a cells were infected with JEV (MOI 0.1), at 1 hpi, cells were treated with DMSO/MTP (10 μM). Cells were harvested at the indicated hpi. (G) Viral RNA levels were quantified using qRT-PCR. Data represents values obtained from two independent experiments ($n = 6$), means ± SD, unpaired Student t-test. (H) Cells were treated as described above. Negative strand of viral RNA was quantified by qRT-PCR. Data is plotted from minimum four independent experiments, expressed as means ± SEM, paired Student t-test. (I) Virus titers was determined by plaque assay. Data represents values obtained from two independent experiments ($n = 6$), means ± SD, unpaired Student t-test. (J–M) Cortical neurons (MOI 1) (J,K), Neuro2a (MOI 1) (L,M), were mock/JEV infected and at 1 hpi, treated with DMSO/MTP (10 μM) (cortical neurons)/ TFP (10 μM) (Neuro2a). (J,L) Cells were harvested at the indicated hpi and viral RNA levels were quantified using qRT-PCR. Data represents values obtained from two independent experiments ($n = 6$). (K,M) Virus titers were measured in culture supernatants using plaque assay, values plotted from two independent experiments ($n = 6$), means ± SD, unpaired Student t-test. Data information: *$P < 0.05$; **$P < 0.01$; ***$P < 0.001$; ****$P < 0.0001$; ns, non-significant. Source data are available online for this figure.

from same intracellular pools and most likely they act on same $Ca^{2+}$ handling channel/pump. Indeed, literature suggests that a variety of phenothiazines including TFP can directly inhibit SERCA pump (Khan et al, 2000). Taken together, our data shows that just like many other phenothiazines, MTP inhibits SERCA pump and induces an increase in cytosolic $Ca^{2+}$ levels. Further, this rise in intracellular $Ca^{2+}$ concentration can activate adaptive ER stress and autophagy. In future, it would be interesting to investigate precise molecular mechanism through which phenothiazines inhibit SERCA pumps as it would be relevant for several other disorders associated with SERCA hyperactivity.

MTP treatment resulted in mild eIF2α phosphorylation and transcriptional activation of ATF4, CHOP, ATF6, Xbp1, and Xbp(s) indicating activation of ER stress. However, no CHOP protein was detectable in drug-treated cells suggesting that the stress was primarily adaptive and not apoptotic. In cells exposed to ER stress, autophagy is transcriptionally activated as a survival response (Avivar-Valderas et al, 2011; Kroemer et al, 2010; Matsumoto et al, 2013; Ogata et al, 2006). Activation of CHOP is mediated by ATF4 and ATF6 (Fusakio et al, 2016; Okada et al, 2002) and generally leads the cells towards apoptosis through upregulation of BH3 only proteins and downregulation of Bcl2 (Matsumoto et al, 2013; McCullough et al, 2001). CHOP also drives inflammation through secretion of IL-1β through caspase-11/caspase-1 (Endo et al, 2006), and activation of NFkB (Willy et al, 2015), and can negatively impact cholesterol and lipid biosynthesis pathways (Chikka et al, 2013). Studies have shown that in response to aa starvation/tunicamycin treatment, the autophagy genes Atg16l1, Map1lc3b, Atg12, Atg3, Beclin1, and Gabarapl2, can be activated by ATF4 independently of CHOP (B'Chir et al, 2013; Rouschop et al, 2010). The autophagy gene activation profile induced by MTP in our study was also similar, along with upregulation of the pro-survival gene Bcl2. Our data indicates that MTP induces a unique chronic/adaptive ER stress with gene expression profiles that are qualitatively distinct from those induced by severe stress such as Thapsigargin.

In the context of JEV, viral RNA translation, viral RNA replication, and particle production are interdependent. Through a careful time-point assessment and polysome analysis, we observe

that phenothiazines do not alter viral RNA translation, but exert a strong autophagy-dependent antiviral effect on the replication complex. This is also supported by significantly reduced levels of JEV negative-strand RNA. Our earlier studies have established that viral nonstructural proteins/replication complex are not targeted for degradation by canonical autophagy (Sharma et al, 2014). It is plausible that autophagy upregulation by phenothiazine reduces the level of LC3-I or another host-factor(s) essential for the virus replication complex biogenesis, maintenance and/or expansion, or alternately targets the stability of the replication complex through recruitment of autophagy-dependent antiviral factors.

The phenothiazines have been shown to inhibit cytokine secretion from microglial cells in animal models of traumatic brain injury, subarachnoid/intracerebral hemorrhage, hypoxia-ischemic recovery etc. (Chen et al, 2019; Dibaj et al, 2012; Fenn et al, 2015; Labuzek et al, 2005; Xu et al, 2017; Zhang et al, 2020; Zhao et al, 2016; Zhou et al, 2019). Patients with schizophrenia and first episode psychosis (FEP) display abnormal profiles of proinflammatory cytokines (especially IL-6 and TNF-α) prior to start of treatment. In these individuals, antipsychotic treatment resulted in decreased serum concentrations of IL-1β, IL-6, IFN-γ, TNF-α, and showed therapeutic effects by reducing microglial inflammation comparable to levels in healthy controls (Marcinowicz et al, 2021). Indeed, we observed a similar effect of the drugs in inflammatory cytokine release from all cell types: primary astrocytes, mixed glial cells, primary cortical neurons, and BMDMs. Degradation of NLRP3 and inhibition of inflammatory cytokine secretion was reversed in Atg5 deficient microglial cells.

TFP and fluphenazine have been shown to be direct inhibitors of TLR3-IRF3 signaling pathway (Zhu et al, 2010). In our study MTP also caused significant inhibition of type I IFN and several other ISGs, indicating suppression of innate immune responses. This was not due to reduced virus replication, as a similar inhibition was also seen in response to LPS treatment. While type I IFN is primarily antiviral in nature, it might also exert a proinflammatory function.

It is also well established that several antipsychotics can cause metabolic disorders such as hypertriglyceridemia, glucose dysregulation and elevated cholesterol levels, which is attributed to transcriptional activation of cholesterol and fatty acid biosynthetic

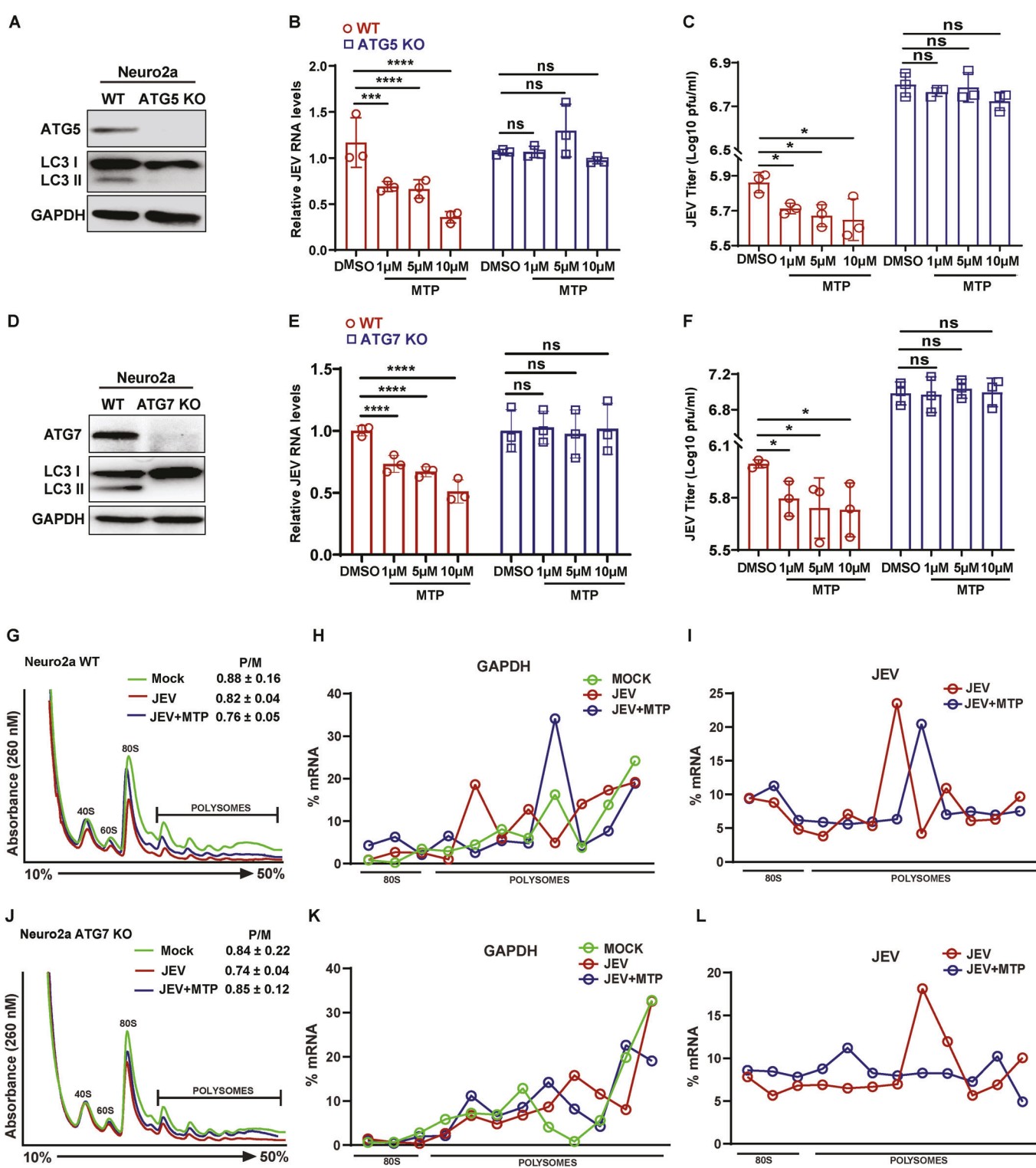

genes via SREBP1 and SREBP2 (Cai et al, 2015; Ferno et al, 2005; Ferno et al, 2006; Pillinger et al, 2020). We too observed that MTP treatment resulted in transcriptional upregulation of several cholesterol biosynthetic genes in divergent cell types. An inverse relation between type I IFN response and flux through the mevalonate pathway has been reported earlier (Blanc et al, 2011;

York et al, 2015). MTP-mediated enhancement of sterol biosynthesis could be directly responsible for the observed downmodulation of type I IFN and inflammation, however, this requires further validation.

In conclusion, our study provides evidence that the phenothiazines MTP and TFP have robust antiviral and neuroprotective

**Figure 7. Antiviral effect of MTP is autophagy-dependent, but does not impact viral RNA translation.**

Neuro2a ATG5 KO and ATG7 knockout cell lines were generated by CRISPR-Cas9. (A) Protein lysates of WT and ATG5KO cells analyzed by immunoblotting using ATG5, LC3 (autophagy control), and GAPDH (internal controls) antibodies. (B,C) Neuro2a WT and ATG5 KO cells were infected with JEV (MOI 1). At 1 hpi, cells were treated with MTP at indicated concentrations. Cells were harvested at 24 hpi to measure JEV RNA levels. (B) Relative viral RNA levels after normalization to respective DMSO-treated control. Data is plotted from three independent experiments ($n = 6$). (C) Culture supernatants were harvested and virus titers was measured using plaque assay, Data is plotted from three independent experiments ($n = 9$). (D) Protein lysates of WT and ATG7 KO cells analyzed by immunoblotting using ATG7, LC3 (autophagy control), and GAPDH (internal controls) antibodies. (E,F) Neuro2a WT and ATG7 KO cells were infected with JEV (MOI 1). At 1 hpi, cells were treated with MTP at indicated concentrations. Cells were harvested at 24 hpi to measure JEV RNA levels. (E) Relative viral RNA levels after normalization to respective DMSO-treated control. Data is plotted from three independent experiments ($n = 9$). (F) Culture supernatants were harvested and virus titers were measured using plaque assay, Data is plotted from three independent experiments ($n = 9$). (G–L) WT and ATG7KO Neuro2a cells were mock/JEV (5 MOI) infected for 1 h, followed by DMSO/MTP (10 µm) treatment till 6 hpi. Cell lysates were prepared from WT (G) and ATG7 KO cells (J), and global polysome profile analysis was performed by the density gradient fractionation system. Polysome-to-monosome (P/M) ratios from two independent experiments are indicated in the respective panel, means ± SD. Percentage distribution of GAPDH mRNA (internal control) (H,K) and viral RNA (I,L) in the monosome and polysome gradient fractions was quantified by qRT-PCR. Similar trends were seen in two independent experiments. Data information: All data are expressed as means ± SD, one-way ANOVA followed by Dunnett test, $*P < 0.05$; $**P < 0.01$; $***P < 0.001$; $****P < 0.0001$; ns, non-significant (H,I,K,L). Source data are available online for this figure.

effects in JE disease condition and have the potential to be repurposed for treatment. These drugs are approved for chronic use and have a high therapeutic index. As a future scope of this work, a small investigator-initiated clinical trial of this drug in JEV patients can test the findings of this pre-clinical study and establish proof of concept in humans.

# Methods

## Ethics statement

All animal experiments were approved by the Institutional Animal Ethics Committee of the Regional Centre for Biotechnology (RCB/IAEC/2018/039). Experiments were performed as per the guidelines of the Committee for the Purpose of Control and Supervision of Experiments on Animals (CPCSEA), Government of India. Animals were kept under a 14 h light/10 h dark cycle at 19–26 °C with ~30–70% humidity. Food and water supply were provided ad libitum. All experiments with JEV were performed in a Biosafety Level 2 laboratory, as per the Biosafety guidelines issued by Department of Biotechnology, Ministry of Science & Technology, Government of India. All in vitro and animal experiments were duly approved by the Institutional Biosafety Committee (RCB/IBSC/18-19/129).

## Cell lines and virus

Neuro2a (mouse neuroblastoma), C6/36 (insect), and Vero cell lines were obtained from the cell repository at the National Centre for Cell Sciences Pune, India. WT and ATG5 KO Mouse embryonic fibroblasts (MEFs) were obtained through the RIKEN Bio-Resource Cell Bank (RCB2710 and RCB2711). WT and ATG5 KO HeLa cell lines were a kind gift from Dr. Richard J. Youle (NIH, USA); and mouse microglia N9 cell line was a gift from Prof. Anirban Basu (NBRC, India). All cell lines were negative for mycoplasma.

Neuro2a cells stably expressing EGFP-TFEB/EGFP-LC3 were generated through plasmid transfection, growth in G418 selection media followed by single cell isolation through FACS. MEFs stably expressing GFP-LC3-RFP-LC3ΔG (Kaizuka et al, 2016), were generated through retroviral transduction as described earlier (Prajapat et al, 2022).

Dulbecco's modified Eagle's medium (DMEM) was used to culture Neuro2a, MEFs, and HeLa cells, Eagle's minimal essential medium (MEM) for Vero cells, RPMI for N9 cells, and Leibovitz's (L-15) medium was used to culture C6/36 cells. All media were additionally supplemented with 10% fetal bovine serum (FBS), 100 µg/ml penicillin-streptomycin, and 2 mM L-glutamine.

JEV isolate Vellore P20778 strain (GenBank accession no. AF080251) was generated in C6/36 cell line. UV-inactivated JEV was generated by exposure of virus to UV ($1600 \times 100 \, \mu J/cm^2$) for 20 min on ice. For animal experiments the mouse-adapted JEV-S3 strain was used (Tripathi et al, 2021). Virus titration was performed in Vero cells using plaque assays as described earlier (Sehrawat et al, 2021).

All reagents, antibodies, and plasmids used in the study are listed in Appendix Table S3.

## Establishment of ATG5 & ATG7 knockout Neuro2a cell lines

ATG5 and ATG7 knockout cell lines were generated by utilizing a CRISPR-Cas9 editing system using pSpCas9(BB)-2A-Puro(PX459) V2.0 (Addgene, #62988) and pSpCas9(BB)-2A-GFP (PX458) (Addgene, #48138) vectors, according to the protocol mentioned previously (Ran et al, 2013). The guide RNA (gRNA) target gene sequences viz. mouse *Atg5* 5'-TTCCATGAGTTTCCGATTGA-3' and mouse *Atg7* 5'-GAACGAGTACCGCCTGGACG-3' cloned into PX459 and PX458 vectors, respectively, were procured from Genescript. The final constructs were named as PX459-gRNA-Atg5 and PX458-gRNA-Atg7. Neuro2a cells were transfected with 2 µg of plasmids using Lipofectamine™ 3000 (Invitrogen™) following the manufacturer's instructions. The empty vectors PX459 and PX458 were used as a respective control. At 48 hours post-transfection, knock out cells were selected either by treating with puromycin (3 µg/ml) for 8 days or by sorting high-intensity GFP-positive cells using a cell sorter BD FACS Aria™ III instrument (BD Biosciences, CA). The knockouts were validated by western blotting.

## Primary cell culture

### Bone marrow-derived macrophage (BMDMs)
BMDMs were isolated from 6–7-week-old C57BL/6 mice. Briefly, mice were euthanized and femurs were dissected, washed with

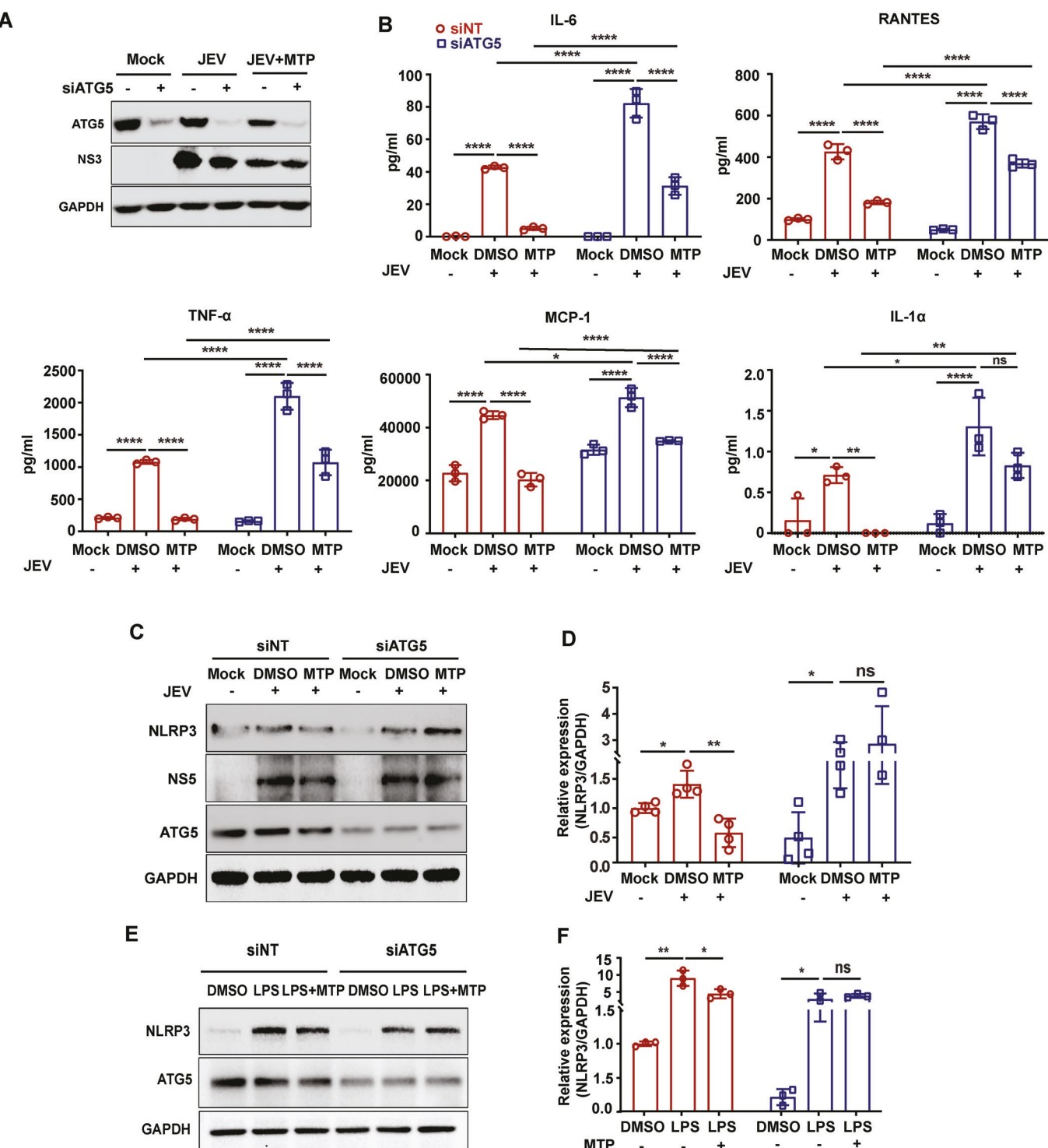

PBS and RPMI media, and flushed with L929-conditioned medium to extrude bone marrow. After RBC lysis, cells were cultured in RPMI complete media supplemented with L929-conditioned media for 7 days. BMDMs were detached using 10 mM EDTA and seeded in 24-well plates for virus infection and drug treatment experiments.

**Embryonic cortical neurons**

Mouse primary cortical neuronal cells were isolated from pregnant mice at embryonic day 16.5 (E16.5). Briefly, embryos were collected by decapitation from pregnant mice, the cortices were dissected from isolated embryonic brains and collected in dissociation media, HBSS (1X sodium pyruvate, 20% glucose, 1 M HEPES, pH 7.3).

**Figure 8.  MTP induced autophagy reduces inflammatory cytokines secretion and NLRP3 levels in microglial cells.**

(A,B) N9 cells were transfected with siNT/ATG5 for 48 h, followed by mock/JEV (MOI 3) infection for 1 h and treatment with MTP (10 μM) till 24 hpi. (A) Cell lysates were prepared and proteins were analyzed by immunoblotting using ATG5, NS3 (infection control), and GAPDH (internal control) antibodies. (B) Culture supernatant was used for the quantitation of cytokine levels using flow cytometry-based CBA assay ($n = 3$). Two-way ANOVA followed by Tukey test. Similar trends were seen in two independent experiments. (C,D) N9 cells were transfected with siNT/ATG5 for 48 h, then infected with JEV (MOI 3) for 1 h. Post-infection, cells were treated with either DMSO or MTP (10 μM) for 24 h. (C) Cell lysates were analyzed by immunoblotting using NLRP3, ATG5, NS5 (infection control), and GAPDH (internal control) antibodies. (D) Bar-graph shows relative expression of NLRP3/GAPDH normalized to mock control from four independent experiments. Unpaired Students t-test. (E,F) siNT/siATG5 transfected cells were treated with DMSO/LPS (1 μg/ml)/LPS (1 μg/ml) +MTP (10 μM) for 24 h. (E) Cell lysates were analyzed by immunoblotting using NLRP3, ATG5, and GAPDH (internal control) antibodies. (F) Bar-graph shows relative expression of NLRP3/GAPDH normalized to DMSO control from three independent experiments. Unpaired Students t-test. Data information: All data are expressed as means ± SD. *$P < 0.05$; **$P < 0.01$; ***$P < 0.001$; ****$P < 0.0001$; ns; non-significant. Source data are available online for this figure.

Tissues were digested with trypsin and DNAse I to make single-cell suspensions. Cells were washed, centrifuged, and resuspended in complete neurobasal medium supplemented with 10% FBS, 20% glucose, 1X Sodium pyruvate, and antibiotics. Finally, cortical neuronal cells were plated on poly-l-lysine coated plates, and media was changed with maintenance media (neurobasal B-27, 1X glutamine, penicillin-streptomycin solution) after every 2 days by adding half new maintenance media.

### Mixed glial culture

Mixed glial cell culture was established as described earlier (Guler et al, 2021). Briefly, P2 pups were decapitated, and their brains were collected in ice-cold 1X HBSS. After careful removal of the meninges, the brain tissue was sliced, treated with 0.05% DNase I in 1X HBSS (40 units per 3 pups), and gently triturated using a 1 ml pipette and a 200 μL pipette, respectively. The mixture was then incubated with 0.05% trypsin for 20 min at RT. Following centrifugation, the cell pellet was suspended in DMEM complete media (2 ml of media for 3 pups). The cells were gently mixed and then passed through a cell strainer to obtain a single-cell suspension. After centrifugation, the pellet was resuspended in 10 ml of complete media, and transferred to T-75 flask coated with poly-l-lysine. Cells were monitored for growth, and media was changed on day 1, 2, and 7. After 10 days, the astrocyte/microglial population was confirmed by immunofluorescence using GFAP and IBA1 antibodies.

### Astrocytes

Astrocytes were separated from mixed glial cell culture as detailed previously (Guler et al, 2021). The oligodendrocyte progenitor cells (OPCs) were detached from confluent mixed glial cell culture through gentle tapping and aspiration. The remaining OPCs were removed with 0.05% DNase I treatment in media for 5 min followed by horizontal shaking of the flask. The adherent astrocyte-microglial layer was washed, trypsinized, and plated onto a sterile bacterial culture plate to allow microglial adherence for 20 min. The astrocyte pool (non-adherent cells), was carefully collected and transferred into a T-75 flask coated with poly-l-lysine. Cells were monitored for growth for 13 days. The purity of isolated astrocytes was confirmed by immunofluorescence using GFAP antibody.

## Virus infection and cell treatment

All virus infection studies were performed by giving mock/JEV infection at indicated MOI for 1 h. Cells were then washed and complete medium was added. Drug treatment was given by adding 10 μM drug/DMSO (control) to mock/JEV-infected cells, which

was maintained till the end of the experiment as indicated. Cells were harvested for cell viability assays, RNA isolation, or western blotting. Culture supernatants were used for quantitative estimation of cytokines using Cytokines bead array (CBA), or virus titers through plaque assay. For autophagy flux and lysosome pH assays, cells were treated with DMSO/Torin1 (1 μM)/MTP (10 μM)/TFP (10 μM) for 6 h. siRNA treatment was performed using mouse-specific Atg5/Atg7/non-targeting (NT) siRNA (50 nM, ON-TARGET plus SMART pool) using the transfection reagent Dharmafect 2 for Neuro2a cells, and Lipofectamine™ RNAimax for MEFs and N9 cells. At 48 h post-transfection, cells were harvested and the knockdown efficiency was measured using western blotting or qRT-PCR. LPS treatment was given at 1 μg/ml for indicated times. Every experiment had biological triplicates and was performed two or more times.

## RNA isolation and quantitative real time (qRT)-PCR

RNA was extracted using Trizol reagent. cDNA was prepared using ImProm-II™ Reverse Transcription System kit, and used to set up qRT-PCR on the QuantStudio 6 (Applied Biosystems). JEV RNA level was determined by specific Taqman probes, and GAPDH was used as an internal control. The gene expression for autophagy, UPR, innate immunity, and lipid/cholesterol biosynthesis was performed with SYBR green reagents. The expression of each gene was calculated by normalization to respective mock/DMSO controls. Each experiment had biological triplicates, and qPCR for each sample was done in technical duplicates or triplicates. The primer sequences for all the genes tested in the study are listed in Appendix Table S4.

## Quantitative analysis of negative sense JEV RNA strand

The negative sense RNA of JEV was quantified using strand-specific RT-qPCR assay, as described previously (Vashist et al, 2012). Total RNA was isolated from JEV-infected Neuro2a cells (MOI 1) treated with either DMSO (control) or 10 μM drug (MTP), and was subjected to strand-specific qRT-PCR. Briefly, cDNA was synthesized using tagged (non-viral sequence) reverse transcription (RT) primer NVnegVneg primer (Appendix Table S4) and real time qPCR was performed using the TaqMan probe (Negative sense_p-robe) selected for the capsid gene (Appendix Table S4), combination of primers that binds to non-viral tag sequence (qPCR-F) and viral strand (qPCR-R) (Appendix Table S4) and the *Premix Ex Taq™* (Probe qPCR) master mix. The absolute amount of negative-sense JEV RNA was determined by the copy numbers extrapolated from the standard curve. The standard curve for negative-strand

RNA was generated using in vitro transcribed minus strand of JEV-capsid gene. A curve was then plotted between Ct values obtained from the range of known RNA concentrations (g/μl) and the copy number (molecules) of negative strand JEV RNA was thus calculated (Faye et al, 2013).

## Western blotting

Cells were lysed in cell lysis buffer (150 mM NaCl, 1% Triton X-100, 50 mM Tris-HCl pH 7.5, 1 mM PMSF, and protease inhibitor cocktail) for 45 min at 4 °C. The supernatant was used for the estimation of protein concentration using BCA assay kit. Cell lysates were mixed with 4X Laemmli buffer (40% glycerol, 20% β-mercaptoethanol, 0.04% bromophenol blue, 6% SDS, 0.25 M Tris-HCl pH 6.8) and boiled at 95 °C for 10 min to denature proteins. Equal concentrations of cell lysates were separated by SDS-PAGE and transferred to PVDF membranes for immunoblotting. The blots were visualized using a Gel Doc XR+ gel documentation system (Bio-Rad) and the expression of proteins was calculated by measuring the intensity of bands using ImageJ (NIH, USA) software. The fold change was calculated after normalization with respective loading controls. All western blotting experiments were performed three or more times, and representative blots are shown.

## Cytokines bead array (CBA)

Mouse microglia N9 cells (biological triplicates) were mock/JEV-infected at indicated MOIs for 1 h. At 12 hpi, cells were treated with DMSO/drugs for 24 h. Alternately N9 cells were treated with 1 μg/ml LPS for 24 h. Supernatants were harvested and used to quantitate the levels of cytokine IL-6, TNF-α, MCP-1, RANTES, IL-1β, and IL-1α using CBA assay as per manufacturer's instruction. The analysis was performed with FCPA array software and the concentration of each cytokine was determined based on their standard curve. All CBA assays were performed two or more times, and representative data from one experiment is shown.

## Animal experiments

The mouse-adapted strain JEV-S3 was generated in 3–4-day-old C57BL/6 mice pups as described earlier (Tripathi et al, 2021). Briefly, pups were infected with JEV ($10^5$ PFU) by an intracranial route. By 3–4 dpi, symptoms of JEV infection such as movement impairment and constant shivering/body tremors were observed. The pups were sacrificed, and the brain tissues were harvested, homogenized in incomplete MEM media, and the supernatant containing infectious virus was titrated by plaque assay.

For JEV survival and other experiments, 3-week-old C57BL/6 mice of either sex were weighed and randomly divided into mock/drug/JEV/JEV+drug groups. Mice were infected intra-peritoneally with $10^7$ PFU JEV, while the mock-infected group received equal volume of incomplete DMEM media by intraperitoneal injection. At 4 hpi, mice were treated with vehicle control (PEG 400) or the drugs MTP (2 mg/kg)/Flubendazole (5 mg/kg)/Fluoxetine (5 mg/kg)/Rilmenidine (5 mg/kg)/TFP (1 mg/kg) in PEG 400 formulations by oral route every 24 h till 15 days. Mice were monitored for symptoms of JEV infection such as change in body weight, movement restrictions, piloerection, tremor, body stiffening, hind

limb paralysis, and mortality. These experiments were not blinded to the investigators.

To determine cytokine levels in brain, mice ($n = 3$) were sacrificed from each group on days 3, 4, 5, 6, and 7, and brain tissues were collected. These were homogenized in lysis buffer, and 30 μg of protein from each sample was used for quantitation of cytokines using LEGENDPLEX MU anti-virus response panel 13-plex assay kit as per manufacturer's instructions. Data was analyzed with LEGENDplex™ Multiplex assay software and the concentration of each cytokine was calculated based on their standard curve.

### Evans blue leakage assay

Mice were intra-peritoneally injected with 100 μl of 2% Evans blue at 3 and 6 dpi. After 45 min they were sacrificed and brains were harvested. Images were captured to visualize the distribution of dye in the brain. For dye quantification, the tissues were weighed and homogenized in dimethylfumarate (DMF) (200 mg/500 μl DMF). The homogenate was heated at 60 °C overnight to ensure complete extraction of the dye. The samples were then centrifuged and absorbance of each sample was measured at 620 nm. Evans blue content was quantitated using a standard curve.

## Cell viability assays

Cell viability assays were performed using the CellTiter-Glo® assay kit as per manufacturer's instructions. The percentage of cell viability was calculated as: [(ATP luminescence for experimental condition)/(ATP luminescence for untreated condition)] × 100 and normalized to mock-infected/untreated DMSO-treated control. MTT assay was performed as described earlier (Sehrawat et al, 2021). LDH assay using culture supernatant from siNT/Atg5 treated microglial cells was performed using CyQUANT™ LDH Cytotoxicity Assay kit as per manufacturer's instructions. The percentage cell death was normalized to siNT mock-infected control.

## Immunofluorescence studies

### Image-based high content screening for virus replication complex

Neuro2a cells were seeded in 96-well black polystyrene microplates (Corning, CLS3603), and infected with JEV at MOI 5. At 1 hpi cells were treated with DMSO/drugs for 24 h, fixed with 4% paraformaldehyde (PFA), and permeabilized with 0.3% Tween-20 for 30 min at RT. After blocking with 5% BSA, JEV-NS1 primary antibody was added (1 h), followed by Alexa Fluor-labeled specific secondary antibody (1 h), and final incubation with DAPI (0.5 μg/ml) for 15 min. Each experiment had one plate with biological triplicates (3 wells) for each condition (mock, JEV + DMSO, JEV +drug). Images were acquired from the entire well area (16 fields per well) on ImageXpress Micro Confocal High-Content Imaging System (Molecular Devices, USA) using FITC and DAPI channels with a ×10 objective lens. The percentage of JEV-NS1 positive cells was calculated based on both DAPI and FITC positive cells using the multi-wavelength cell scoring module of the MetaXpress software.

### TFEB nuclear translocation assay

EGFP-TFEB stable Neuo2a cells were treated with DMSO/Torin1 (1 μM)/MTP (10 μM)/TFP (10 μM) for 6 h, fixed, and stained with

DAPI, and images were acquired on the High-Content imaging system as described above. Each plate had biological triplicates for all conditions. The Pearson coefficient between GFP and DAPI was calculated using the translocation module of the MetaXpress software (cut off = 0.5). The percentage of EGFP-TFEB nuclear translocation per well was calculated using Torin1 as a positive control.

### Autophagy flux assay

The flux assay was performed as described earlier (Prajapat et al, 2022). MEFs stably expressing GFP-LC3-RFP-LC3ΔG were seeded at a density of 10,000 cells/well in 96-well black polystyrene microplates, followed by treatment of DMSO/Torin1 (1 μM)/Baf A1 (100 nM)/MTP (10 μM)/TFP (10 μM) for 6 h. Cells were fixed with 4% PFA and stained with DAPI. Images were recorded from 16 fields per well that covered the entire well area, on ImageXpress Micro Confocal High-Content Imaging System using DAPI, FITC, and Texas red channels with a ×10 objective lens. Analysis of images was performed using multi-wavelength cell scoring module of the MetaXpress software, that calculates the integral intensity of GFP and RFP from triple positive (DAPI, GFP, RFP) cells. GFP/RFP ratio was estimated and normalized to DMSO-treated control. For autophagy flux inducer, the cut-off value of GFP/RFP was set to <0.8, while GFP/RFP > 1.2 was considered for autophagy flux inhibitors. Torin1 (GFP/RFP = 0.59) and BafA1 (GFP/RFP = 1.25) were used as positive and negative controls, respectively.

### GFP-LC3 puncta formation

Neuro2a cells stably expressing EGFP-LC3 were grown on glass coverslips. These were treated with DMSO/Torin1 (1 μM)/MTP (10 μM)/TFP (10 μM) for 6 h, fixed with 4% PFA, and mounted using ProLong Gold anti-fade reagent with DAPI. Images were acquired by Elyra PS1 (Carl Zeiss Super-resolution microscope) with ×60 objective (lasers 405, 488 nm). LC3 puncta were counted from 20 cells acquired from two independent coverslips using 'Analyse particles' plugin algorithm of ImageJ (Fiji).

## Endosome acidification assay

LysoTracker Red and LysoSensor Yellow-Blue assays were performed as described earlier (Albrecht et al, 2020; Prajapat et al, 2022) Briefly, Neuro2a cells were grown on glass coverslips, and treated with DMSO (control)/Torin1 (1 μM)/BafA1 (100 nM) or MTP/TFP (10 μM) for 6 h, followed by incubation with 10 μM LysoTracker red (40 min), or 10 μM LysoSensor Yellow-Blue (5 min). Cells were then washed with ice-cold PBS three times and fixed using 4% PFA. Imaging was done on LSM 880 microscope, Carl Zeiss. LysoSensor Yellow-Blue imaging was done using the excitation wavelength range of 371–405 nm and emission wavelength range of 420–650 nm. The LysoSensor dye has dual-emission peaks of 440 nm (blue in less acidic organelles) and 540 nm (yellow in more acidic organelles). The analysis of LysoTracker red, and LysoSensor Yellow-Blue (yellow) fluorescence intensities were performed using ImageJ (Fiji) and normalized to DMSO-treated control, from 50 cells across two independent coverslips.

## Measurement of oxidative stress

Neuro2a cells were mock/JEV (MOI 5) infected for 1 h, followed by DMSO/FDA drug (10 μM) treatment till 24 hpi. DMF (70 μM) and

NAC (3 mM) were added at 16 hpi and maintained till 24 hpi. Post-treatment, cells were incubated with 5 mM of oxidative stress indicator CM-H2DCFDA in incomplete media for 15 min. Cells were washed with PBS and fluorescence intensity was measured using flow cytometry BD FACS Verse (BD Biosciences, USA). All FCS files were analyzed via FlowJo software and represented as mean fluorescence intensity.

## Virus entry assay through qRT-PCR

JEV entry can be quantitatively measured by estimating endocytosed virus levels through qRT-PCR at 1 hpi as described previously (Sehrawat et al, 2021). Briefly, Neuro2a cells were pre-treated with DMSO/MTP (10 μM)/TFP (10 μM) for 1 h, and then infected with JEV (MOI 5) for 1 h in the presence of the drug. Cells were harvested by washing, and trypsin treatment was given to remove any extracellular attached virus. qRT-PCR was done to measure the levels of internalized viral RNA relative to the GAPDH transcript.

## Immunofluoresence-based virus entry assay

Neuro2a cells stably expressing EGFP-LC3 were grown on glass coverslips. These were mock/JEV (MOI 50) infected, and DMSO/MTP (10 μM) was added at 1 hpi and maintained for another 1 h. The cells were then fixed with 4% PFA and immunostained for the JEV envelope antibody. Coverslips were mounted using ProLong Gold anti-fade reagent with DAPI, and images were acquired by Elyra PS1 (Carl Zeiss Super-resolution microscope) with ×63 objective (lasers 405, 488, and 561 nm). Colocalization between EGFP-LC3 and JEV-envelope was calculated by individual Spots and Spot to Spot colocalization per cell using Imaris 8 software.

## Calcium imaging

Calcium imaging was performed as reported earlier (Arora et al, 2021; Tanwar et al, 2022). Neuro2a, primary astrocytes, MEFs were cultured on confocal dishes (SPL Life Sciences, 200350) to attain 60–80% confluency. Cells were incubated in culture medium containing 4 μM fura-2AM for 45 min at 37 °C. Post-incubations, cells were washed 3 times and bathed in HEPES-buffered saline solution (2 mM CaCl₂, 1.13 mM MgCl₂, 140 mM NaCl, 10 mM D-glucose, 4.7 mM KCl and 10 mM HEPES; pH 7.4) for 5 min. Further, 3 washes were given and cells were bathed in HEPES-buffered saline solution without 2 mM CaCl₂ to ensure removal of extracellular Ca²⁺ before starting the measurements. A digital fluorescence imaging system (Nikon Eclipse Ti2 microscope coupled with CoolLED pE-340 Fura light source and a high-speed PCO camera) was used, and fluorescence images of several cells were recorded and analyzed. Excitation wavelengths—340 nm and 380 nm—were used alternately for Fura-2AM and emission signal was recorded at 510 nm. Ca²⁺ traces represent average data from multiple cells from a single imaging dish (number of cells is denoted by "n" on the graphs). Bar graphs represent data from multiple imaging experiments. The number of cells and traces for different conditions are specified on the respective bars as n = x, y where x stands total number of cells imaged and y means total number of independent experiments performed.

## The paper explained

### Problem

Japanese encephalitis (JE) is an arboviral disease, and the leading global cause of viral encephalitis. It is endemic in south-east Asia, and several cases are reported every year mainly in the pediatric population. Due to global warming, and diversification of mosquito habitats, recent cases have also been reported in areas such as mainland Australia. These highlight the universal threat of JE expansion. The disease causes nearly 30% mortality, and of the survivors, ~50% develop permanent neurological complications. Treatment mostly aims to relieve symptoms and support body functions, with no effective antiviral therapy available. Overall there is an urgent need to supplement existing treatment strategy.

### Results

This study aimed to identify and characterize drugs with repurposing potential for JE. Based on our previous work, we curated a pool of FDA-approved drugs with autophagy-inducing potential, and tested these for both antiviral and anti-inflammatory effects in cell lines, and in the mouse model of JE. The antipsychotic phenothiazine drug Methotrimeprazine (MTP) significantly improved JE infected mice survival, reduced neuroinvasion and provided protection against blood-brain barrier breach and neuroinflammation. Another widely prescribed antipsychotic phenothiazine, Trifluoperazine (TFP) showed similar antiviral and neuroprotective effects in vitro and in vivo. Mechanistically, MTP caused dysregulation of ER $Ca^{2+}$ homeostasis, and induced a unique adaptive ER stress signature, resulting in upregulation of autophagy flux. This exerted a dual antiviral and neuroprotective effect.

### Impact

Phenothiazine drugs are widely used in clinical practice for the treatment of bipolar disorders, psychosis and schizophrenia, and can reach the brain/CNS which is an added advantage for infections such as JE. They are approved for chronic use and have a high therapeutic index with well-tolerated side-effects, and can be administered to pediatric patients. These thus have potential for a paradigm shift in the treatment of JE.

## Polysome profiling

Polysome profiling was performed following a previously reported protocol (Kumara et al, 2023), with minor modifications. Cell pellets were resuspended in 500 μl of lysis buffer A (50 mM HEPES pH 7.4, 10 mM KOAc, 20 mM Mg(OAc)$_2$, 70 mM sucrose, 5 mM DTT, 1X protease inhibitor cocktail (Roche), RNase inhibitor 5 U/ml, 1 mM PMSF, 100 μg/ml cycloheximide, 200 μg/ml Heparin and 1% Triton X-100). Cells were lysed using a Dounce glass homogenizer by applying 50 strokes on ice. The cell lysate was centrifuged at 15,000 rpm for 30 min to pellet the cell debris. 10 OD units of each supernatant were layered on a 10–50% sucrose gradient in buffer B (20 mM HEPES pH-7.4, 150 mM KOAc, 20 mM Mg(OAc)$_2$, 5 mM DTT, 100 μg/ml cycloheximide). The sucrose gradients were prepared using the Gradient Master from Biocomp Instruments, and centrifuged at 38,000 rpm, 4 °C for 3 h in Beckman Ultracentrifuge (rotor SW41). The gradients were fractionated in the Piston Gradient Fractionator from Biocomp Instruments. 0.3 ml of each fraction was collected and stored at −80 °C for further analysis. RNA was isolated starting from fractions corresponding to the 80S subunit and beyond. After

cDNA preparation, qRT-PCR was performed for JEV RNA and GAPDH as described above. Using the CT values, the percent (%) distribution of the mRNA across the gradient was calculated using the ΔCT as described (Panda et al, 2017).

## Statistical analysis

None of the mice were excluded from the experiments. No blinding was done during the experiments. Statistical analysis was performed using paired/unpaired Student's t-test, one-way ANOVA followed by Dunnett test/two-way ANOVA followed by Tukey test, and Log-rank (Matel-Cox) test. Differences were considered significant at $P$ values of *$p < 0.05$, **$p < 0.01$, ***$p < 0.001$, and ****$p < 0.0001$, as indicated in the figures. Error bar indicates means ± SD/SEM. All graphs were plotted and analyzed using GraphPad Prism 8.

## For more information

For more information, see https://www.who.int/news-room/fact-sheets/detail/japanese-encephalitis, https://www.cdc.gov/japaneseencephalitis/index.html, https://www.ecdc.europa.eu/en/japanese-encephalitis, and https://ncvbdc.mohfw.gov.in/index1.php?lang=1&level=1&sublinkid=5773&lid=3693.

## Data availability

This study includes no data deposited in external repositories.

## Peer review information

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

## Acknowledgements

This work was supported by DBT grant BT/PR27875/Med/29/1302/2018 to MK. RKM is supported by the DBT/Wellcome Trust India Alliance Intermediate Fellowship (IA/I/19/2/504651). PSK acknowledges funding from SERB grant (CRG/2022/002656). SV would like to acknowledge grant no. JCB/2021/000015 from the Science and Engineering Research Board, and grant no. BT/MED/32/11/2019 from the Department of Biotechnology, Govt. of India. The Small Animal Facility is supported by DBT grant BT/PR5480/INF/158/2012. The funders had no role in the study design, data collection, and interpretation, or the decision to submit the work for publication. EC is supported by SERB-National post-doctoral fellowship (PDF/2021/001436). SKP, LM, KA, and SC are supported by DBT fellowship, SK, PS, and NK are supported by UGC fellowship. We thank Padmakar Tambare and the staff of the Small Animal Facility for their help with animal experiments. We also acknowledge facilities and staff of Advanced Technology Platform Centre. We are grateful to Dr. Kiran Bala Sharma for critical inputs on the manuscript. All

Virology lab members are acknowledged for useful discussions and support. The graphical abstract was produced using bioRender.

## Author contributions

**Surendra K Prajapat**: Data curation; Formal analysis; Validation; Investigation; Visualization; Methodology; Writing—original draft; Writing—review and editing. **Laxmi Mishra**: Data curation; Investigation; Methodology. **Sakshi Khera**: Data curation; Investigation; Methodology. **Shadrack D Owusu**: Data curation; Investigation; Methodology. **Kriti Ahuja**: Data curation; Investigation; Methodology. **Puja Sharma**: Data curation; Investigation; Methodology. **Eira Choudhary**: Conceptualization; Data curation; Formal analysis; Validation; Investigation; Methodology; Writing—review and editing. **Simran Chhabra**: Data curation; Formal analysis; Investigation; Methodology. **Niraj Kumar**: Data curation; Investigation; Methodology. **Rajan Singh**: Methodology. **Prem S Kaushal**: Conceptualization; Resources; Formal analysis; Supervision; Methodology; Writing—review and editing. **Dinesh Mahajan**: Conceptualization; Methodology; Writing—review and editing. **Arup Banerjee**: Investigation; Methodology. **Rajender K Motiani**: Conceptualization; Resources; Formal analysis; Supervision; Methodology; Writing—original draft; Writing—review and editing. **Sudhanshu Vrati**: Resources; Project administration. **Manjula Kalia**: Conceptualization; Resources; Formal analysis; Supervision; Funding acquisition; Methodology; Writing—original draft; Project administration; Writing—review and editing.

## Disclosure and competing interests statement

The authors declare no competing interests.

# Expanded View Figures

**Figure EV1.  MTP inhibits the secretion of proinflammatory cytokines from JEV infected/LPS-stimulated astrocytes.**

(A) Primary astrocytes were isolated from P2 pups, and purity was confirmed through immunofluorescence staining with GFAP antibody. Scale bar, 10 μm. (B) Primary astrocytes were treated with DMSO/MTP (10 μM)/LPS (1 μg/ml)/LPS + MTP for 24 h. Percentage cell viability was measured by MTT assay ($n = 3$). (C,D) Primary astrocytes were mock/JEV (MOI 1) infected for 1 h, followed by treatment with either DMSO or MTP (10 μM) till 24 h. (C) Viral RNA levels were quantified using qRT-PCR. Graph shows the relative expression levels of JEV RNA normalized to DMSO-treated control. Data is plotted from two independent experiments ($n = 6$). (D) Culture supernatant was used to determine virus titers using plaque assays. Data represents values obtained from two independent experiments ($n = 6$). (E,F) Primary astrocytes were infected with JEV (MOI 1) for 1 h then treated with DMSO/MTP (10 μM) till 24 h (E), or were treated with DMSO (mock)/LPS (1 μg/ml)/LPS + MTP for 24 h (F). Culture supernatants were collected and cytokine levels were quantitated by CBA using flow cytometry. Data were analyzed with LEGENDplexTM Multiplex assay software. Data shows values from one representative experiment ($n = 3$). Similar trends were seen in two independent experiments. Data information: All data expressed as means ± SD, statistical significance was determined using unpaired Student t-test. *$P < 0.05$; **$P < 0.01$; ***$P < 0.001$; ****$P < 0.0001$.

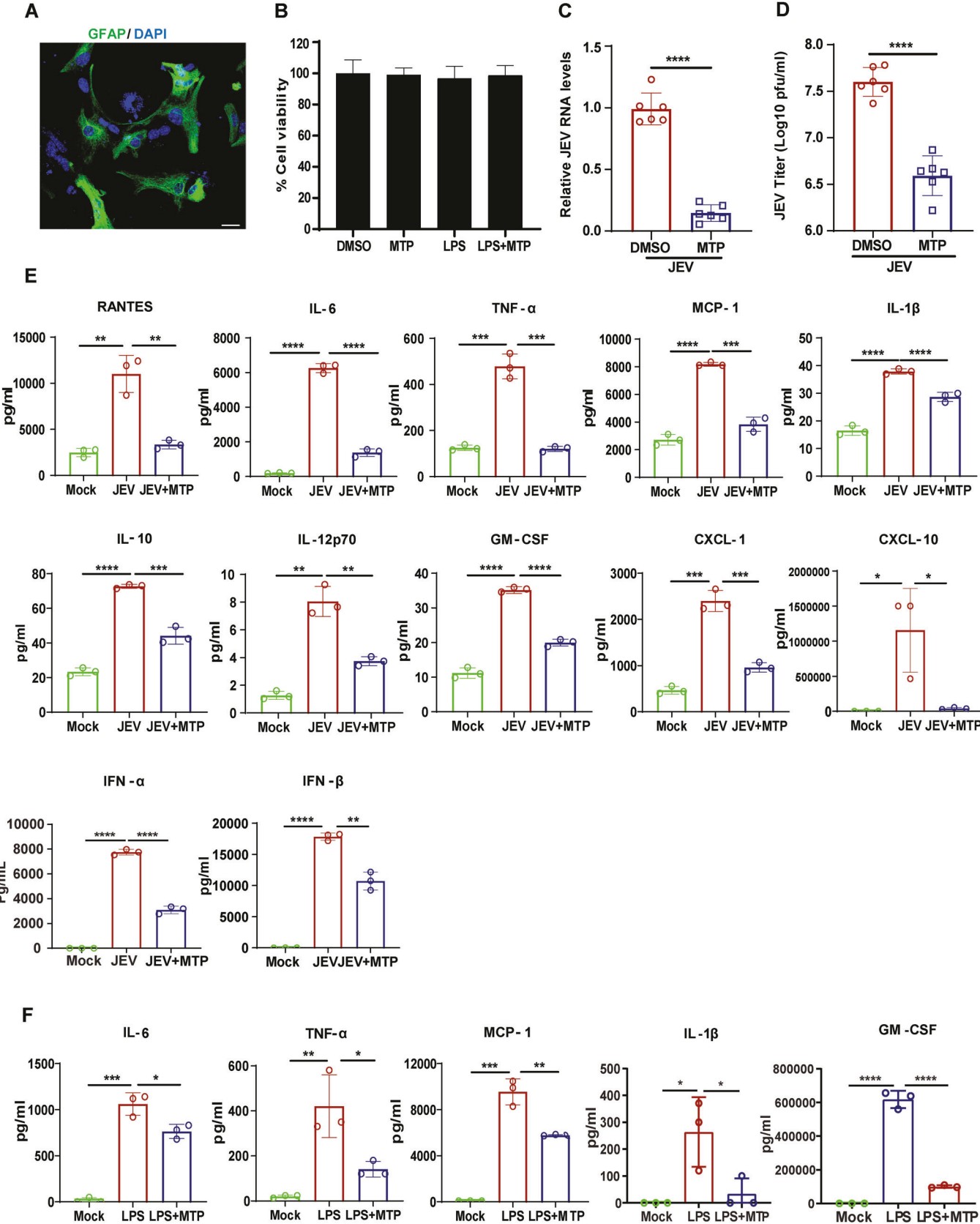

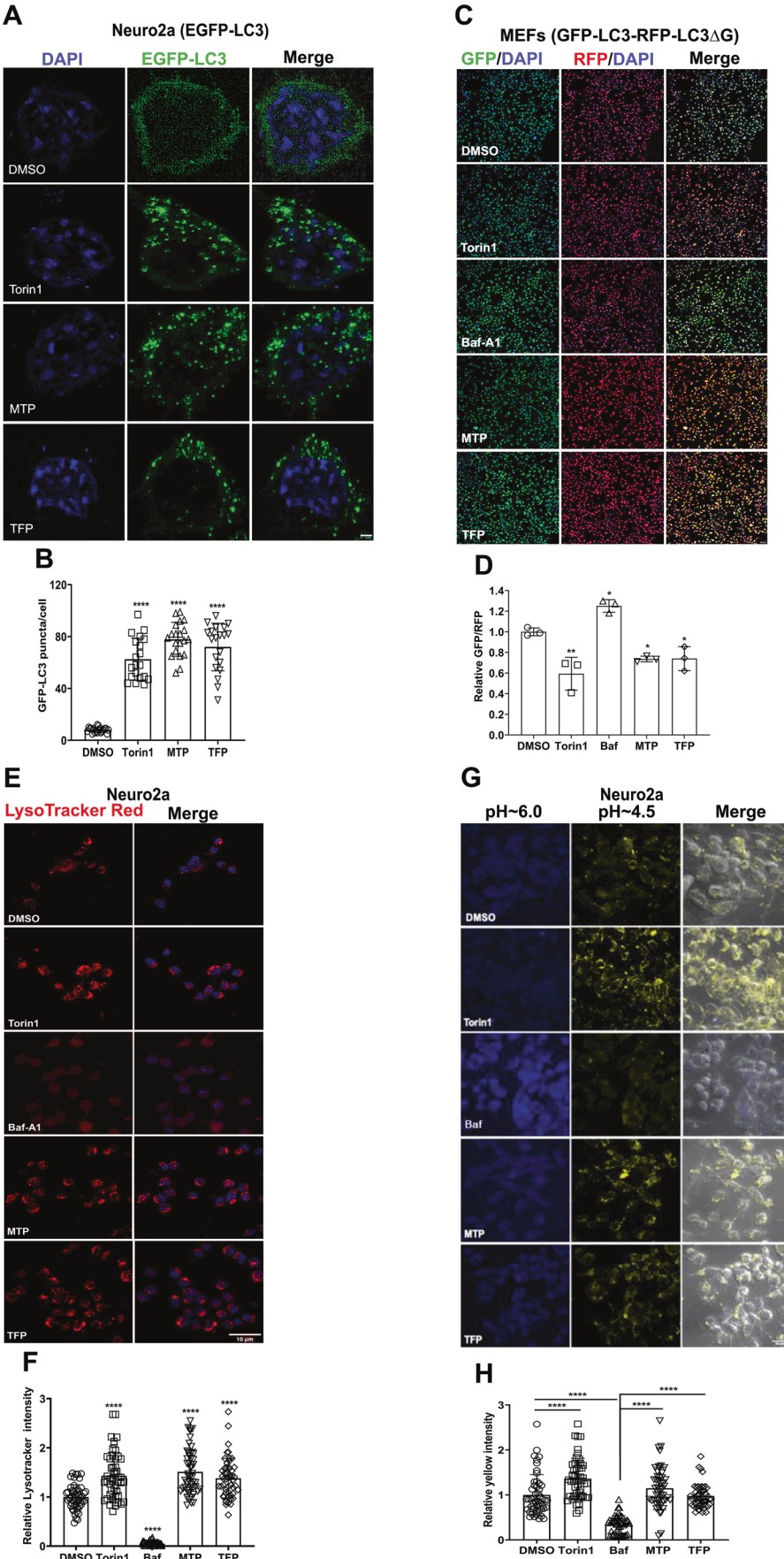

◀ **Figure EV2. Phenothiazines induce functional autophagy flux and do not alter lysosomal pH.**

(A,B) EGFP-LC3 expressing stable Neuro2a cells were treated with DMSO/Torin1 (1 μM)/MTP (10 μM)/TFP (10 μM) for 6 h. (**A**) Representative SIM images are shown. Scale bar, 10 μm. (**B**) Bar-graph shows quantitation of EGFP-LC3 puncta per cell. Data is acquired from 20 cells across two independent coverslips. (**C,D**) GFP-LC3-RFP-LC3ΔG expressing stable MEFs were treated with DMSO (control)/Torin1 (1 μM)/BafA1 (100 nM) or MTP/TFP (10 μM) for 6 h. (**C**) Images were visualized by high-content imaging system. Scale bar, 100 μm. (**D**) Graph showing GFP/RFP ratios ($n = 3$). (**E–H**) Neuro2a cells grown on glass coverslips were treated with indicated drugs as described above for 6 h, followed by incubation with 10 μM LysoTracker Red for 40 min (**E,F**) or 10 μM LysoSensor Yellow-Blue for 5 min (**G,H**). Representative confocal images are shown. Scale bar, 10 μm (**E**); 20 μm (**G**). LysoTracker Red (**F**) and LysoSensor Yellow-Blue (yellow) (**H**) fluorescence intensities were calculated from 50 cells across two independent coverslips using ImageJ (Fiji). Data information: All data are normalized to DMSO control and expressed as means ± SD, one-way ANOVA test followed by Dunnett test was used for statistical significance. *$P < 0.05$; **$P < 0.01$; ****$P < 0.0001$.

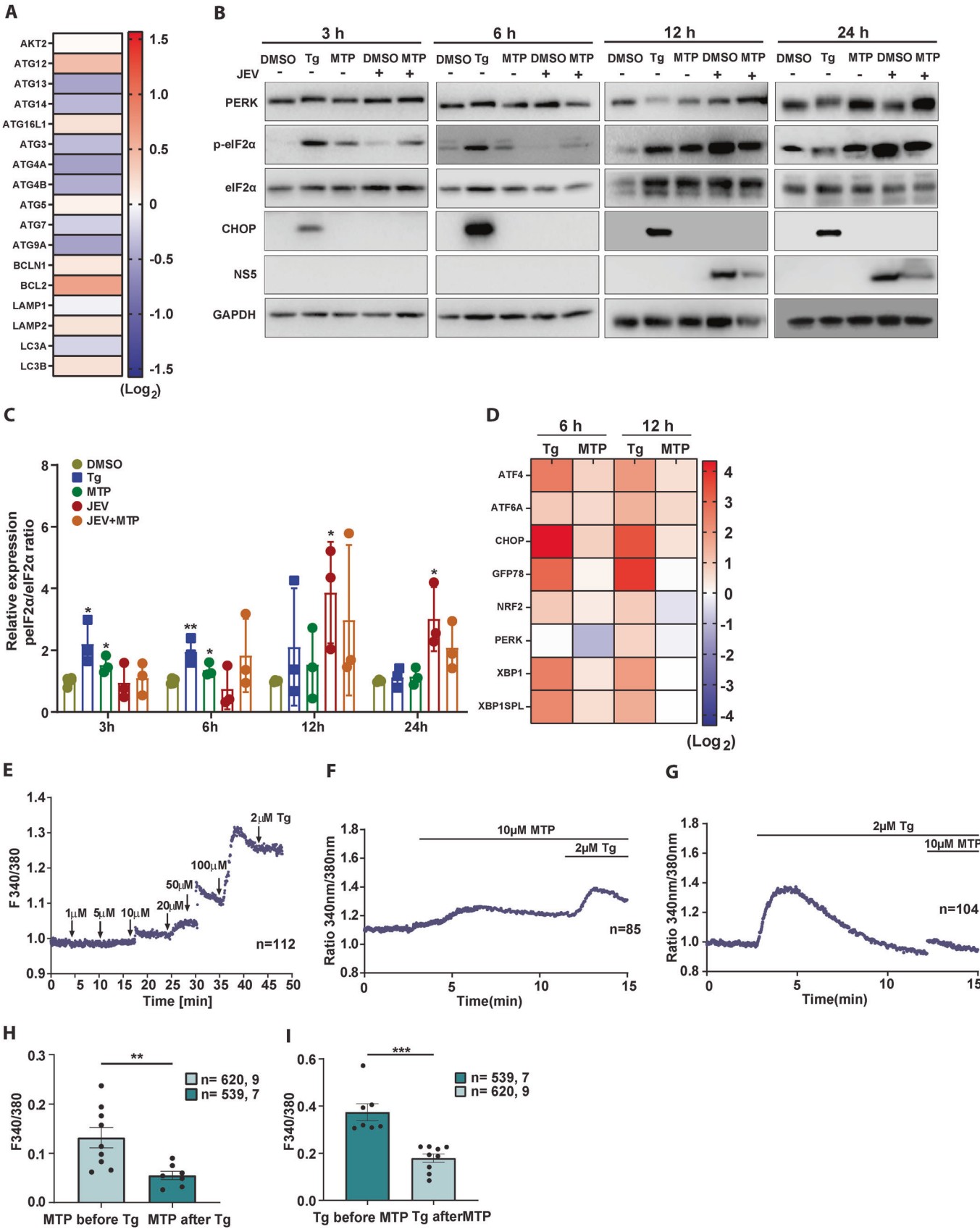

◄  **Figure EV3.  MTP activates adaptive ER stress and dysregulates ER calcium homeostasis.**

(A) MEFs were treated with DMSO/MTP (10 µM) for 6 h and mRNA levels of autophagy genes were determined by qRT-PCR. Heatmap shows relative gene expression level after normalization to DMSO-treated control ($n = 3$). (B,C) MEFs were infected with JEV (MOI 1), at 1 hpi cells were treated with DMSO/Tg (1 µM)/MTP (10 µM) for the indicated time points. (B) Protein lysates were analyzed by immunoblotting using PERK, p-eIF2α, eIF2α, CHOP, NS5 (infection control) and GAPDH (loading control) antibodies. (C) Bar-graph shows relative expression of p-eIF2α/eIF2α normalized to DMSO control from three independent experiments, unpaired Student t-test. (D) MEFs were treated with DMSO/Tg (1 µM)/MTP (10 µM) for the indicated time points. mRNA levels of ER stress markers and chaperones were quantified using qRT-PCR. Heatmap depicts relative gene expression normalized to DMSO control, represented as mean ($n = 3$). (E) Representative Ca$^{2+}$ imaging trace of MTP dose–response assay, where "$n = 112$" denotes the number of cells in that particular trace. Cells were stimulated with increasing doses of MTP- 1 µM, 5 µM, 10 µM, 20 µM, 50 µM and 100 µM followed by addition of 2 µM thapsigargin (Tg) in Ca$^{2+}$-free buffer. (F) Representative Ca$^{2+}$ imaging trace of experiments where cells were stimulated with 10 µM MTP in absence of extracellular Ca$^{2+}$ followed by addition of 2 µM Tg. Here, "$n = 85$" denotes the number of cells in that particular trace. (G) Representative Ca$^{2+}$ imaging trace of experiments where cells were stimulated first with 2 µM Tg to deplete ER Ca$^{2+}$ stores, followed by addition of 10 µM MTP in absence of extracellular Ca$^{2+}$. Here, "$n = 104$" denotes the number of cells in that particular trace. (H) Quantitation of MTP (10 µM) induced ER Ca$^{2+}$ stores depletion before and after the addition of 2 µM Tg. 620 and 539 cells from 9 and 7 independent imaging dishes were analyzed for the two conditions, respectively. (I) Quantitation of Tg, (2 µM) induced ER Ca$^{2+}$ stores depletion before and after the addition of 10 µM MTP. 539 and 620 cells from 7 and 9 independent imaging dishes were analyzed for the two conditions, respectively ("$n = x, y$" where "$x$" denotes total number of cells imaged and "$y$" denotes number of traces recorded). Data presented are mean ± S.E.M., unpaired Student's $t$ test, **$P < 0.01$; ***$P < 0.001$.

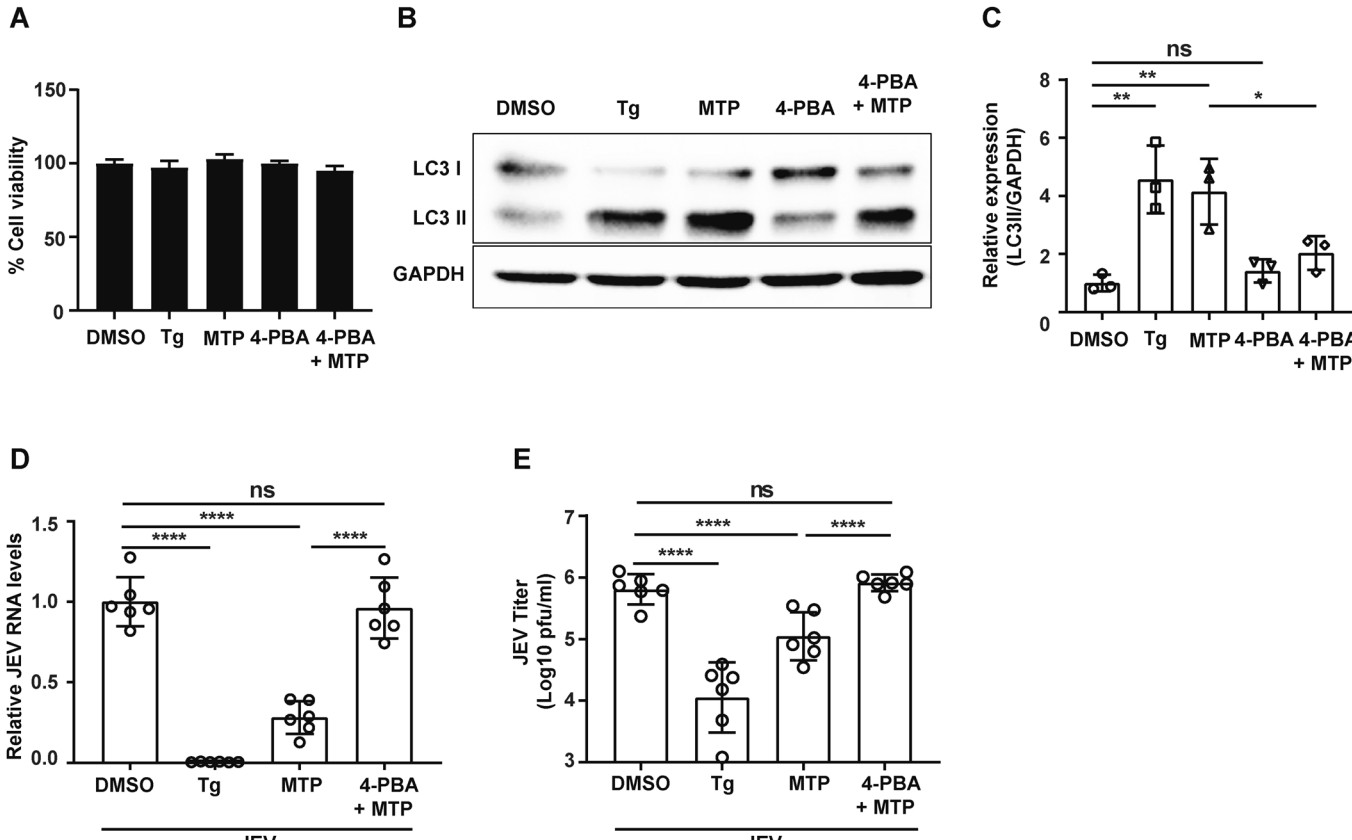

**Figure EV4. MTP induced ER stress is essential for autophagy and antiviral effect.**

(A) Neuro2a cells were treated with DMSO/Tg (1 μM)/MTP (10 μM)/4-PBA (2 mM)/4-PBA (2 mM) + MTP (10 μM) for 24 h. MTT assay was used to calculate % cell viability, and normalized to DMSO treated control ($n = 3$). (B,C) Neuro2a cells were treated with DMSO/Tg (1 μM)/MTP (10 μM)/4-PBA (2 mM)/4-PBA (2 mM) + MTP (10 μM) for 6 h. (B) Protein lysates were analyzed by immunoblotting using LC3 and GAPDH (loading control) antibodies. (C) Bar-graph shows relative protein expression level of LC3II/GAPDH calculated after normalization to DMSO control. Values were plotted from three independent experiments. (D,E) Neuro2a cells were infected with JEV at MOI 1 for 1 h. Post-infection, cells were treated with DMSO/Tg (1 μM)/MTP (10 μM)/4-PBA (2 mM) + MTP (10 μM) for 24 h. (D) Cells were harvested, viral transcript levels were measured using qRT-PCR and normalized to DMSO-treated infected control from two independent experiments ($n = 6$). (E) Virus titers was measured in culture supernatant using plaque assay, value plotted from two independent experiments ($n = 6$). Data information: All data were expressed as means ± SD, unpaired Student t-test was used to calculate statistical significance *$P < 0.05$; **$P < 0.01$; ***$P < 0.001$; ****$P < 0.0001$; ns, not significant.

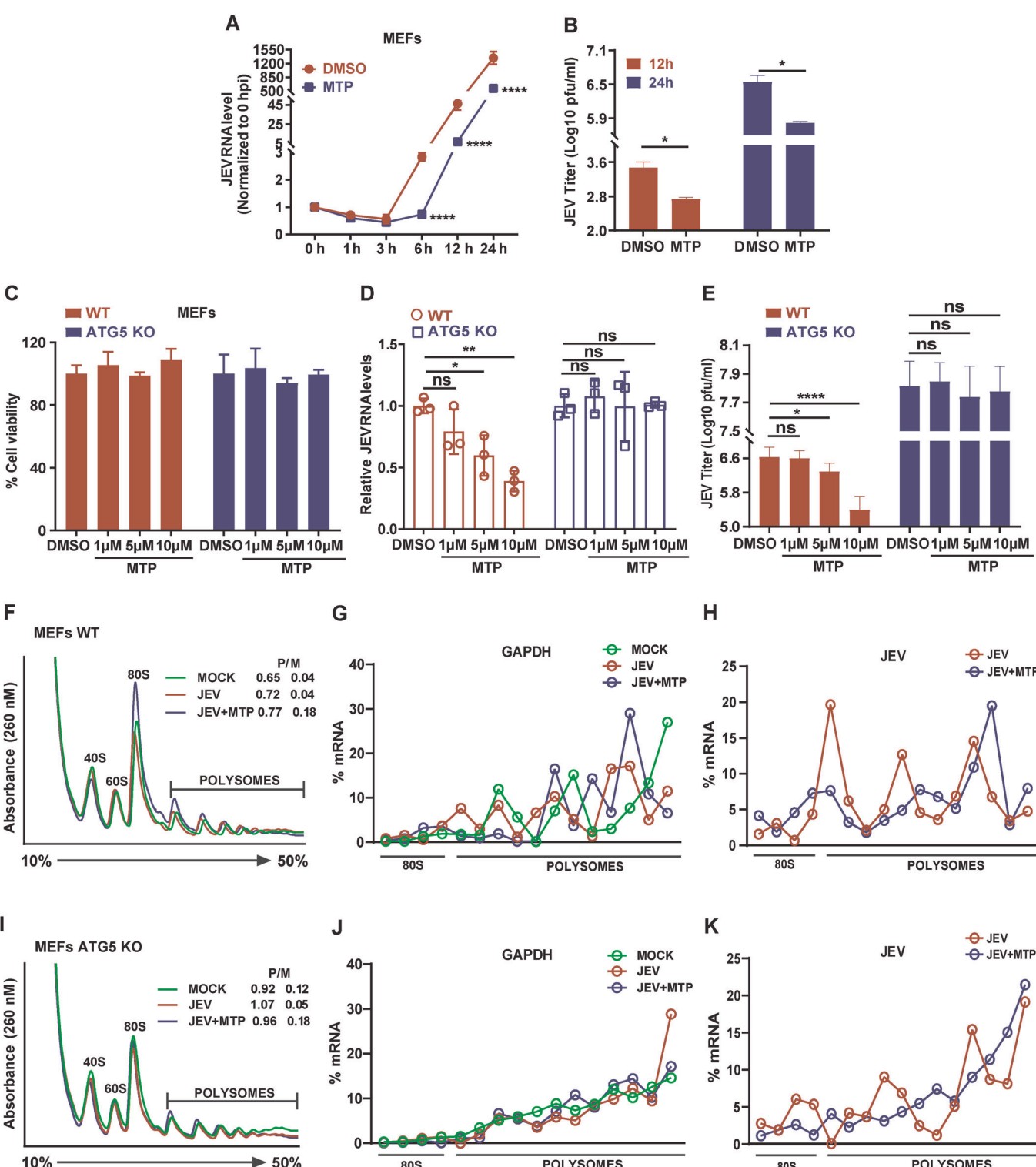

◄ **Figure EV5. Antiviral effect of phenothiazines is autophagy dependent.**

(A,B) MEFs were mock/JEV infected (MOI 1) and at 1 hpi, treated with DMSO/MTP (10 μM). (A) Cells were harvested at the indicated hpi and viral RNA levels were quantified using qRT-PCR. Data represents values obtained from two independent experiments ($n = 6$). (B) Culture supernatant was used to determine virus titers using plaque assay. Data is plotted from two independent experiments ($n = 6$), and compared by unpaired Student t-test. (C) WT and ATG5 KO MEFs were treated with indicated concentrations of MTP for 24 h, and the percentage cell viability was measured and normalized to respective DMSO-treated controls ($n = 3$). (D,E) WT and ATG5 KO MEFs were infected with JEV at MOI 1, and at 1 hpi treated with MTP at indicated concentrations. Cells were harvested at 24 hpi and the relative viral RNA levels were quantitated using qRT-PCR, and plotted after normalization to respective DMSO-treated control. Data represents values from three independent experiments ($n = 9$). (E) Culture supernatant was collected and virus titers was determined using plaque assay. Data is plotted from three independent experiments ($n = 9$). (F–K) WT and ATG5 KO MEFs were mock/JEV (5 MOI) infected for 1 h, followed by DMSO/MTP (10 μm) treatment till 6 hpi. (F,I) Global polysome profile analysis of cell lysates were performed by the density gradient fractionation system. Polysome-to-monosome (P/M) ratios from two independent experiments, means ± SD. (G,H,J,K) Percentage distribution of GAPDH mRNA (housekeeping gene) (G,J), viral RNA (H,K) in the monosome and polysome fractions was analyzed by qRT-PCR. Similar trends were seen in two independent experiments. Data information: One-way ANOVA followed by Dunnett test was used for the determination of statistical significance, *$P < 0.05$; **$P < 0.01$; ****$P < 0.0001$, ns, non-significant.

