## [Peer Review File · EMBO Molecular Medicine]

Methotrimeprazine is a neuroprotective antiviral in JEV infection via adaptive ER stress & autophagy

Surendra Prajapat, Laxmi Mishra, Sakshi Khera, Shadrack Owusu, Kriti Ahuja, Puja Sharma, Eira Choudhary, Simran Chhabra, Niraj Kumar, Rajan Singh, Prem S. Kaushal, Dinesh Mahajan, Arup Banerjee, Rajender Motiani, Sudhanshu Vrat, and Manjula Kalia

DOI: [10.15252/emmm.202317813](https://doi.org/10.15252/emmm.202317813)

Corresponding author: Manjula Kalia (manjula@rcb.res.in)

Review Timeline:

Submission Date:	5th Apr 23
Editorial Decision:	26th May 23
Appeal:	30th May 23
Editorial Decision:	6th Jun 23
Revision Received:	1st Nov 23
Editorial Decision:	17th Nov 23
Revision Received:	24th Nov 23
Accepted:	24th Nov 23

Editor: Zeljko Durdevic

Transaction Report:

26th May 2023

Decision on your manuscript EMM-2023-17813

Dear Dr. Kalia,

Thank you for the submission of your manuscript to EMBO Molecular Medicine, and please accept my apologies for the delay in getting back to you. We have now received feedback from two of the three reviewers who agreed to evaluate your manuscript. As the referee #3 will unfortunately not be able to return his/her report in a timely manner, and given that both reviewers provide very similar recommendations, we prefer to make a decision now in order to avoid further delay in the process.

As you will see from their reports pasted below, while they recognize interest of your study, they also raise serious concerns, particularly regarding, but not limited to, the the lack of proper controls, the unclear mechanistic insight, the lack of validation of the findings in an adequate animal model and inadequate statistical analyses. As clear and conclusive insight into a novel, clinically relevant observation is crucial for publication in EMBO Molecular Medicine, and together with the fact that we only accept papers that receive enthusiastic support upon initial review, I am afraid that we cannot offer to consider the manuscript further.

I am sorry that I could not bring better news this time and hope that the referee comments are helpful in your continued work in this area.

Yours sincerely,

Zeljko Durdevic

***** Reviewer's comments *****

Referee #1 (Remarks for Author):

This study investigates the protective role of the autophagy inducer antipsychotic phenothiazines methotrimeprazine (MTP) and trifluoperazine (TFP) both in vivo and in vitro during Japanese encephalitis virus (JEV) infection. The authors linked drug-induced autophagy and anti-inflammatory/antiviral effect and concluded that MTP and TFP have robust antiviral and neuroprotective effects in JE disease, which has certain significance for the prevention and control of JE. There is an interesting work. Some data to support the authors' claim are reasonable, however, the reviewer has several concerns especially mechanism-related authors' claim in the paper as follows.

1. Astrocytes are another major source of proinflammatory cytokines in JEV-induced CNS inflammation; why were microglia used in the initial screening of anti-inflammatory drugs?
2. Is the downregulation of pro-inflammatory factors in 1C-1F due to drug-inhibited replication of JEV?
3. In result 2, MTP-treated mice inhibited viral replication, which may account for the downregulation of proinflammatory cytokines. The downregulation of type I IFN (IFN- α with IFN-B) in MTP-treated mice further supported this finding. Therefore, result 2 does not directly prove the anti-inflammatory effect of phenothiazines in vivo.
4. More tests should be performed in primary cortical neurons, such as changes in P62?
5. In 5J, p-p70S6K/p70S6K appeared to be upregulated in the TFP-treated group compared to the DMSO-treated group?
6. P-ERK usually has two bands detected. Why did only one band appear in this study?
7. GAPDH is inconsistent in some WB results, e.g. 5J, 6A, 6C,7A, etc.

8. What's the meaning of "JEV infected cells showed detectable eIF2 α phosphorylation", in the line 269-270? As Fig 7A shows, detectable eIF2 α phosphorylation occurred in the absence of JEV infection.
9. It is recommended that ATG5 and ATG7 knockout N2a cell lines be constructed and backfilled for further validation.
10. In the JEV (MOI 3) group in Figure S8A, why is it on two membranes?
11. In Figure S4, does IL- β refer to IL-1 β ?
12. In result 5, "Phenothiazines induce adaptive ER stress and dysregulation of intracellular calcium signaling" is there a link between changes in calcium signaling and autophagy/anti-inflammatory or autophagy-antiviral? And what is the significance of the result in the context of this study?
13. In Results 6, "MTP induces ER stress and negatively regulates type I interferon signaling in virus infected cells" Is there a link between changes in type I interferon signaling and autophagy-anti-inflammation or autophagy-antiviral? And what is the significance of the result in the context of this study?
14. The exact mechanisms between drug-induced autophagy and anti-inflammation, as well as drug-induced autophagy and anti-viral, need to be sufficiently well resolved.
15. It would be better to provide a schematic diagram.

Referee #2 (Remarks for Author):

Prjapat et al report on studies of the impact of autophagy on JEV infection and neuroinflammation. The authors have already published numerous manuscripts showing that JEV replication is inhibited by autophagy. Here they tested the effects of FDA-approved drugs that induce autophagy in a JEV mouse model and in JEV infected cells in vitro. They show convincingly that two antipsychotic medications, methotrimeprazine and trifluoperazine (MTP and TFP) improve survival in a lethal murine model of JEV encephalitis, with a decrease in viral RNA detected, blood-brain barrier (BBB) disruption, and neuroinflammation. The rest of the manuscript uses neural cell lines and MEFs to interrogate the mechanism of these effects. The data ultimately show a disconnect between autophagy and viral RNA levels, despite the strong title. In addition, the lack of evaluation of primary neural cells and validation of findings in their in vivo model dampens enthusiasm. In addition, many of the key experiments lack appropriate controls and statistical analyses. Below are comments by figure:

Figure 1. The authors need to use standard plaque assays or PCR for both positive and negative strands of JEV RNA to demonstrate the effects of drugs on viral replication. The detection of viral RNA is not the same as showing infectious virions. This is true of all figures that evaluate JEV replication in cells and tissues.

Figure 2. Again, plaque assays are required to demonstrate infectious virions in the CNS at all time-points. Also, it appears that viral entry and not viral replication is impacted by MTP as levels of virus are the same in JEV vs. JEV-MTP by 7 dpi. It is also unclear whether the inhibition of innate immune responses is a positive or negative effect, as >50% of animals die of infection, which could be the result of lack of anti-viral immunity.

Figure 3. Can the authors add similar studies using primary neural cells? Also, multi-step growth curves are needed using standard plaque assays.

Figure 4. Again, the detection of viral RNA does not indicate infectious virions. Please perform multi-step growth curves with results on a log scale. Also, the JEV survival curve in 4I is different from the one depicted in 2A. Please explain.

Figure 5. The authors need to perform all Western blot experiments multiple times and show quantitation with statistical analyses. Note that the level of p62 are not included in the figure.

Figure 6. Please provide information on Thapsigargin and why it is included in this study. It is also difficult to assess the data in this figure without labeling of time-points and quantitation and statistical analyses of multiple blots. Finally, MEFs are irrelevant for understanding the impact of anti-psychotic drugs on JEV infection of the brain. These experiments should be performed in primary astrocytes or microglia. The calcium transient data are not very convincing as the levels induced by MTP are very low. Finally, the authors need to use a SERCA blocker to confirm that these are channels mediating calcium effects.

Figure 7. Please state which cell types are being used in these experiments. Panel A top does not show any differences between results with or without JEV. Please provide quantitation of all Western blotting experiments with statistical analyses. Overall, the data in this figure do not define the mechanisms that underlie the in vivo effects of MTP during JEV encephalitis.

Figure 8. Please use plaque assays to assess viral replication. Also, the siRNA experiments lack siRNA controls.

Text issues:

1. The authors need to define all protein acronyms and introduce their function for readers.
2. Discussion of dopaminergic agonists is not relevant as these are an entirely different class of drugs.

As a service to authors, EMBO provides authors with the possibility to transfer a manuscript that one journal cannot offer to publish to another EMBO publication. The full manuscript and if applicable, reviewers reports are automatically sent to the receiving journal to allow for fast handling and a prompt decision on your manuscript. For more details of this service, and to transfer your manuscript to another EMBO title please click on Link Not Available

Dear Dr. Durdevic

We thank you and the reviewers for the time and efforts with our manuscript review. After carefully going over the reviewers' analysis and critique of our manuscript, we would request you to consider the following submissions.

Regarding lack of proper controls, and inadequate statistical analyses:

Unfortunately, **Reviewer 2 has missed out on several important points** of our study, and we strongly feel that his/her evaluation of our manuscript should be revisited.

1. All the viral RNA quantification has been done using plus strand viral RNA qPCR. Further, **plaque assay data is performed for several** of the experiments (Fig 1 & Fig 2).
2. We have included the **appropriate controls in every experiment such as non-targeting siRNA, and cell viability data.**
3. **Every western blot has been performed 3 or more times** and the quantification is represented for many of the experiments. We can include quantitative data for the remaining blots.
4. All the appropriate statistical tests have been performed.
5. Thapsigargin is a SERCA blocker and the calcium experiments are performed with Thapsigargin as a positive control. These details are included in the manuscript text. Further, the reviewer is comparing MTP mediated ER calcium release with Thapsigargin induced ER calcium depletion and thereby concluding that the MTP stimulated calcium changes are low. Yes, MTP mediated ER calcium release is relatively lower when compared with Thapsigargin. This would be the case with all test molecules acting on ER calcium mobilization machinery because Thapsigargin is a robust and most efficient (among the SERCA blockers reported so far) ER calcium releaser. Thapsigargin would almost completely deplete ER calcium stores due to a very high electrochemical gradient working across ER and cytosol. Therefore, in absolute terms MTP stimulated ER calcium release is significant however in relative (in comparison to Thapsigargin) terms it appears small particularly when MTP and Thapsigargin data is plotted on the same graphs.

Regarding unclear mechanistic insight:

We believe that our study does provide mechanistic insight, however it is possible that the data supporting this may not have been adequately highlighted. We clearly demonstrate that induction of the ER stress response is a primary stimulus for autophagy. We have shown that the mechanism of the antiviral role of autophagy is due to inhibition of viral protein synthesis, which we can further consolidate through additional experiments. The anti-inflammatory effect of autophagy is well documented in the field, which we can corroborate with addition of new data. Additionally, in the revised manuscript, we will include a highly informative diagrammatic summary of the work, which will include all of these mechanistic details.

Regarding the lack of validation of the findings in an adequate animal model:

The mouse model used in the study is a well-established experimental model of JE pathogenesis. Further, we have substantiated our study by doing key experiments in primary mouse cortical neurons and primary bone-marrow derived macrophages. We can extend our study to include primary microglial cells and astrocytes as has been suggested.

We believe that all the points made by both the Reviewers can be adequately addressed in a revised version of the manuscript. If given a chance we can revise the manuscript in the next 3 months. We would greatly appreciate if you could reconsider the decision by going over the comments again and evaluate our responses.

Looking forward to your prompt optimistic response.

Thanking you,

Sincerely

Manjula Kalia

6th Jun 2023

Dear Dr. Kalia,

Thank you for your response to the editorial decision on your manuscript entitled "Methotrimeprazine is a neuroprotective antiviral in JEV infection via adaptive ER stress & autophagy". I have carefully examined the arguments provided in your letter and discussed them with other members of our editorial team.

I am pleased to inform you that we decided to re-consider our initial decision and to invite major revision of your manuscript. Please provide detailed responses to the referee concerns, use only scientific argumentation in response to the referee criticisms and appropriately amend the manuscript to strengthen main message of the study.

Further consideration of a revision that addresses reviewers' concerns in full will entail a second round of review. EMBO Molecular Medicine encourages a single round of revision only and therefore, acceptance or rejection of the manuscript will depend on the completeness of your responses included in the next, final version of the manuscript. As we cannot guarantee the outcome of the second round of review, and to save you from any frustrations in the end, I would strongly advise against returning an incomplete revision.

We would welcome the submission of a revised version within three months for further consideration. Please let us know if you require longer to complete the revision.

I look forward to receiving your revised manuscript.

Yours sincerely,

Zeljko Durdevic

We require:

- 1) A .docx formatted version of the manuscript text (including legends for main figures, EV figures and tables). Please make sure that the changes are highlighted to be clearly visible.
- 2) Individual production quality figure files as .eps, .tif, .jpg (one file per figure). For guidance, download the 'Figure Guide PDF': (<https://www.embopress.org/page/journal/17574684/authorguide#figureformat>).
- 3) A .docx formatted letter INCLUDING the reviewers' reports and your detailed point-by-point responses to their comments. As part of the EMBO Press transparent editorial process, the point-by-point response is part of the Review Process File (RPF), which will be published alongside your paper.
- 4) A complete author checklist, which you can download from our author guidelines (<https://www.embopress.org/page/journal/17574684/authorguide#submissionofrevisions>). Please insert information in the checklist that is also reflected in the manuscript. The completed author checklist will also be part of the RPF.
- 5) Please note that all corresponding authors are required to supply an ORCID ID for their name upon submission of a revised

manuscript.

6) It is mandatory to include a 'Data Availability' section after the Materials and Methods. Before submitting your revision, primary datasets produced in this study need to be deposited in an appropriate public database, and the accession numbers and database listed under 'Data Availability'. Please remember to provide a reviewer password if the datasets are not yet public (see <https://www.embopress.org/page/journal/17574684/authorguide#dataavailability>).

13) Author contributions: You will be asked to provide CRediT (Contributor Role Taxonomy) terms in the submission system. These replace a narrative author contribution section in the manuscript.

14) A Conflict of Interest statement should be provided in the main text.

Please note: When submitting your revision you will be prompted to enter your funding and payment information. This will allow Wiley to send you a quote for the article processing charge (APC) in case of acceptance. This quote takes into account any reduction or fee waivers that you may be eligible for. Authors do not need to pay any fees before their manuscript is accepted and transferred to the publisher.

EMBO Press participates in many Publish and Read agreements that allow authors to publish Open Access with reduced/no publication charges. Check your eligibility: <https://authorservices.wiley.com/author-resources/Journal-Authors/open-access/affiliation-policies-payments/index.html>

Response to Reviewer's comments

We thank the reviewers for their insightful comments. We have addressed these using further experimentation. Several initial observations been validated in primary cell cultures: Astrocytes, mixed glial cultures and cortical neurons (JEV infection and LPS-stimulation). We have generated ATG5 and ATG7 CRISPR ko Neuro2a cell lines, and validated the autophagy dependent antiviral effect. Calcium flux experiments have been extended to Neuro2a cells and primary astrocytes. We also show that rescue of ER-stress via 4-PBA, blocks MTP induced autophagy induction, and antiviral effect, demonstrating the critical upstream role of the ER stress response. Quantification of JEV negative-strand RNA has been performed, which clearly indicates the antiviral effect of MTP at the level of the JEV replication complex. We also include several Polysome profiling experiments showing that MTP does not alter JEV translation. The autophagy dependent role of MTP in degradation of the inflammasome NLRP3 protein has been shown.

New information/modification in the revised manuscript are highlighted in blue font.

Referee #1 (Remarks for Author):

This study investigates the protective role of the autophagy inducer antipsychotic phenothiazines methotrimeprazine (MTP) and trifluoperazine (TFP) both in vivo and in vitro during Japanese encephalitis virus (JEV) infection. The authors linked drug-induced autophagy and anti-inflammatory/antiviral effect and concluded that MTP and TFP have robust antiviral and neuroprotective effects in JE disease, which has certain significance for the prevention and control of JE. There is a interesting work. Some data to support the authors' claim are reasonable, however, the reviewer has several concerns especially mechanism-related authors' claim in the paper as follows.

1. Astrocytes are another major source of proinflammatory cytokines in JEV-induced CNS inflammation; why were microglia used in the initial screening of anti-inflammatory drugs?

Response: As the reviewer has correctly stated, both astrocytes and microglia are a major source of proinflammatory cytokines. For the initial screening, a routinely used and standardized microglial cell line was chosen. We have now validated our observations with MTP in primary astrocytes (**Figure 3**), and primary mixed glial cultures (**Figure EV1**).

2. Is the downregulation of pro-inflammatory factors in 1C-1F due to drug-inhibited replication of JEV?

Response: The drug-induced inhibition of JEV replication in microglial cells (**Appendix Figure S2G**), could contribute partially to down-regulation of pro-inflammatory factors seen in Fig 1C-1F. However, the drug also has a strong independent anti-inflammatory effect as can be seen in LPS treated microglial cells (**Appendix Figure S5**, new data in **Figure 10E-F**), and in LPS treated primary mixed glial (new data in **Figure EV1F**), primary astrocytes (new data in **Fig 3F**), and primary cortical neurons (new data in **Appendix Figure S4D**).

3. In result 2, MTP-treated mice inhibited viral replication, which may account for the downregulation of proinflammatory cytokines. The downregulation of type I IFN (IFN- α with IFN-B) in MTP-treated mice further supported this finding. Therefore, result 2 does not directly prove the anti-inflammatory effect of phenothiazines *in vivo*.

Response: To support our findings that the phenothiazines exert an anti-inflammatory effect *in vivo*, we now show inhibition of pro-inflammatory cytokine release from LPS treated primary astrocytes (new data in **Figure 3F**), primary mixed glial (new data in **Figure EV1F**), and primary cortical neurons (new data in **Appendix FigS4D**). The *in vivo* anti-inflammatory effects of phenothiazines are also documented in literature and discussed in the manuscript (lines 480-486; Ref 79-87).

4. More tests should be performed in primary cortical neurons, such as changes in P62?

Response: We have included several additional experiments with primary cortical neurons (**Appendix Figure S4**). We also show MTP induced changes in p62 levels in primary cortical neurons (**Figure 5G-H**) and time-course of virus infection (**Figure 8B J-K**).

5. In 5J, p-p70S6K/p70S6K appeared to be upregulated in the TFP-treated group compared to the DMSO-treated group?

Response: We now show a different western blot (**Figure 5L**). The quantification of three independent western blots is also presented (**Figure 5M**).

6. P-ERK usually has two bands detected. Why did only one band appear in this study?

Response: The western blotting has been performed using PERK (protein kinase R-like ER kinase) antibody, and not P-ERK (phospho-Extracellular signal related kinase) (**Figure 6B, EV3B**). All acronyms have been expanded in the revised manuscript.

7. GAPDH is inconsistent in some WB results, e.g. 5J, 6A, 6C,7A, etc.

Response: We have provided new western blots for all these figures (**Figure 5L, 6B, EV3B**). We also provide quantification of western blot data from 3-independent experiments.

8. What's the meaning of "JEV infected cells showed detectable eIF2 α phosphorylation", in the line 269-270? As Fig 7A shows, detectable eIF2 α phosphorylation occurred in the absence of JEV infection.

Response: Fig 7A of our original manuscript (upper panel), lacked the Thapsigargin positive control for p-eIF2 α , and we inadvertently showed a higher exposure blot of p-eIF2 α . We now show western blots with the Thapsigargin (positive control) treated cells, where it is evident that the level of p-eIF2 α is undetectable in JEV infected cells at early time points (**Figure EV3B-C**).

9. It is recommended that ATG5 and ATG7 knockout N2a cell lines be constructed and

backfilled for further validation.

Response: We have constructed ATG5 and ATG7 CRISPR knockout Neuro2a cells and validated the autophagy dependent antiviral effect in these cell lines (**Figure 9**).

10. In the JEV (MOI 3) group in Figure S8A, why is it on two membranes?

Response: We now show new blots for the same condition (MOI 3) group (**Figure EV5A**).

11. In Figure S4, does IL- β refer to IL-1 β ?

Response: Yes, corrected to IL-1 β (**Appendix Fig S6E**).

12. In result 5, "Phenothiazines induce adaptive ER stress and dysregulation of intracellular calcium signaling" is there a link between changes in calcium signaling and autophagy anti-inflammatory or autophagy-antiviral? And what is the significance of the result in the context of this study?

Response: Our data suggests that MTP binding results in inhibition of SERCA channels in diverse cell types-neuronal, fibroblasts, primary astrocytes (**Figure 6, EV3, Appendix Fig S8**) and a consequent induction of adaptive ER stress in these cells, which is characterized by a moderate upregulation of ER stress sensors, a marginal enhancement of p-eIF2 α , but no CHOP production. This is the primary stimulus for autophagy upregulation, and new data in our revised manuscript demonstrates that pharmacological rescue of ER stress through 4-PBA completely blocks MTP induced autophagy and reverses the antiviral effect (**Figure 7**).

13. In Results 6, "MTP induces ER stress and negatively regulates type I interferon signaling in virus infected cells" Is there a link between changes in type I interferon signaling and autophagy-anti-inflammation or autophagy-antiviral? And what is the significance of the result in the context of this study?

Response: Our current data does not suggest that changes in type I interferon signalling directly impacts the autophagy mediated antiviral effect, and these are likely to be independent events. However, further studies with type I interferon receptor signalling deficient cells are required to dissect these completely. The MTP treatment mediated antiviral effect is largely at the level of the virus replication complex which is either inhibited or targeted to negatively regulate virus replication. The downregulation of inflammatory signalling, is likely to be mediated by degradation of the inflammasome components through autophagy (**Figure 10C-F**), and this is well documented in several studies (**PMIDs: 21124315; 21151103; 22286270; 269883397; 30966861**). Similarly, autophagy has been shown to negatively regulate type I IFN signalling (**PMID: 17709747**) and also leads to degradation of innate immune sensors (**PMID: 28898289; 32715615**).

14. The exact mechanisms between drug-induced autophagy and anti-inflammation, as well as drug-induced autophagy and anti-viral, need to be sufficiently well resolved.

Response: Our published studies have shown that autophagy is antiviral for JEV, however

viral non-structural proteins, or the replication complex are not targeted through canonical autophagy. Interestingly, non-lipidated LC3-I, marks replication complexes, and is an essential host factor for virus replication (**PMID: 25046112; 33095129**). Here, we have attempted to further dissect the mechanism between drug-induced autophagy and the antiviral effect. We have ruled out a negative impact of autophagy on viral protein translation through polysome profiling (**Figure 9G-L, EV4F-K, Appendix Table S2**). Our data thus indicates that drug-induced autophagy directly reduces virus replication complexes as evidenced by significant decrease of negative-strand replicative intermediate (**Figure 8H**). This could be driven by autophagy dependent modulation (restriction) of an essential host-factor (such as LC3-I), or ER-membranes, that are either required for replication complex biogenesis/maintenance, or through destabilization of the replication complex by recruitment of autophagy dependent antiviral factors.

We also show that drug-induced autophagy is required for degradation of the inflammasome protein NLRP3, both in infected and LPS-stimulated microglial cells (**Figure 10 C-F**). The anti-inflammatory effects of autophagy are also well documented in published literature (**PMIDs: 21124315; 21151103; 22286270; 269883397; 30966861**).

15. It would be better to provide a schematic diagram.

Response: We now provide a schematic diagram to show the different steps of the virus infection process (**Figure 8A**), and a comprehensive schematic summarising the study (**Figure 11**).

Referee #2 (Remarks for Author):

Prjapat et al report on studies of the impact of autophagy on JEV infection and neuroinflammation. The authors have already published numerous manuscripts showing that JEV replication is inhibited by autophagy. Here they tested the effects of FDA-approved drugs that induce autophagy in a JEV mouse model and in JEV infected cells in vitro. They show convincingly that two antipsychotic medications, methotrimeprazine and trifluoperazine (MTP and TFP) improve survival in a lethal murine model of JEV encephalitis, with a decrease in viral RNA detected, blood-brain barrier (BBB) disruption, and neuroinflammation. The rest of the manuscript uses neural cell lines and MEFs to interrogate the mechanism of these effects. The data ultimately show a disconnect between autophagy and viral RNA levels, despite the strong title. In addition, the lack of evaluation of primary neural cells and validation of findings in their in vivo model dampens enthusiasm. In addition, many of the key experiments lack appropriate controls and statistical analyses. Below are comments by figure:

Figure 1. The authors need to use standard plaque assays or PCR for both positive and negative strands of JEV RNA to demonstrate the effects of drugs on viral replication. The detection of viral RNA is not the same as showing infectious virions. This is true of all figures that evaluate JEV replication in cells and tissues.

Response: We have included plaque assay data for all experiments (**Figure 1B, 2C, 3D, 4C, 4E, 4G, 7E, 8I, 8K, 8M, EV1D, EV4B, EV4E, EV5D, Appendix Fig S6C, S9B, S9E, S10C, S10F**). We also show absolute quantification of JEV negative strand RNA to show the inhibitory effect of MTP on JEV replication (**Figure 8H**).

Figure 2. Again, plaque assays are required to demonstrate infectious virions in the CNS at all time-points. Also, it appears that viral entry and not viral replication is impacted by MTP as levels of virus are the same in JEV vs. JEV-MTP by 7 dpi. It is also unclear whether the inhibition of innate immune responses is a positive or negative effect, as >50% of animals die of infection, which could be the result of lack of anti-viral immunity.

Response: JEV plaque assay data from the CNS at all time points has been shown in **Figure 2C**. The levels of virus at 7 dpi do not reflect virus entry, but virus titres in the brain on that day. Using entry assays in both primary cortical neurons and Neuro2a cells (**Figure 8B-C**), we show that MTP does not affect virus entry but rather virus replication. Further, our study suggests that the MTP mediated inhibition of the innate immune and inflammatory response is likely to be a positive effect as 50% of the animals survive the lethal dose of the virus. In contrast, the untreated mice do not survive, despite having very high levels of interferons.

Figure 3. Can the authors add similar studies using primary neural cells? Also, multi-step growth curves are needed using standard plaque assays.

Response: We now show similar data with primary neuronal cells (**Appendix Fig S4**), and multi-step growth curves including plaque assay data (**Figure 8, EV4 A-B, Appendix S9 A-B**).

Figure 4. Again, the detection of viral RNA does not indicate infectious virions. Please perform multi-step growth curves with results on a log scale. Also, the JEV survival curve in 4I is different from the one depicted in 2A. Please explain.

Response: We now show multi-step plaque assay data in **Figure 4C, 4E, 4G**. The JEV survival curve in 4I (now **Figure 4K**) is different from 2A, because the survival experiment with TFP was performed at sub-lethal dose as detailed in the results, figure legend and M&M.

Figure 5. The authors need to perform all Western blot experiments multiple times and show quantitation with statistical analyses. Note that the level of p62 are not included in the figure.

Response: Western blot experiments have been performed three or more times, and we show quantification with statistical analyses. The p62 western blot and the quantification graph in the original manuscript were shown in Figure 5G & H (now **Figure 5I & J**).

Figure 6. Please provide information on Thapsigargin and why it is included in this study. It is also difficult to assess the data in this figure without labeling of time-points and quantitation and statistical analyses of multiple blots. Finally, MEFs are irrelevant for

understanding the impact of anti-psychotic drugs on JEV infection of the brain. These experiments should be performed in primary astrocytes or microglia. The calcium transient data are not very convincing as the levels induced by MTP are very low. Finally, the authors need to use a SERCA blocker to confirm that these are channels mediating calcium effects.

Response: Thapsigargin is a SERCA blocker and a positive control for the ER stress and calcium experiments. We now show quantification of western blots from 3-independent experiments (**Fig 6B-C, EV3 B-C**). We also show adaptive ER stress signatures and calcium flux data in Neuro2a cells (**Figure 6E-H**), and primary astrocytes (**Appendix Fig S8B-E**). The details of the calcium experiments are included in the manuscript text.

Further, we cannot compare MTP mediated ER calcium release with Thapsigargin induced ER calcium depletion and conclude that the MTP stimulated calcium changes are low. Yes, MTP mediated ER calcium release is relatively lower when compared with Thapsigargin in Neuro2a cells and MEFs. This would be the case with most of the test molecules acting on ER calcium mobilization machinery because Thapsigargin is a robust and highly efficient (among the SERCA blockers reported so far) ER calcium release inducer. Thapsigargin would almost completely deplete ER calcium stores due to a very high electrochemical gradient working across ER and cytosol. Therefore, in absolute terms MTP stimulated ER calcium release is significant however in relative (in comparison to Thapsigargin) terms it appears small particularly when MTP and Thapsigargin data is plotted on the same graphs. Importantly, in primary astrocytes MTP inhibits SERCA almost comparable to Thapsigargin thereby highlighting that MTP is a potent and versatile SERCA inhibitor (**Appendix Figure S8B-E**).

Using an ER stress inhibitor (4-PBA), we show that the induction of autophagy and the subsequent antiviral effect is entirely dependent on the activation of ER stress responses (**Figure 7**). The data from Figure 6 of the original version, which was based on MEFs, have now been moved to **Fig EV3E-G**, along with labelling of time-points, quantification and statistical analyses. The ER stress and calcium responses are now also shown in both Neuro2a cells (**Figure 6E-H**), and in primary astrocytes (**Appendix Fig S8B-E**).

Figure 7. Please state which cell types are being used in these experiments. Panel A top does not show any differences between results with or without JEV. Please provide quantitation of all Western blotting experiments with statistical analyses. Overall, the data in this figure do not define the mechanisms that underlie the in vivo effects of MTP during JEV encephalitis.

Response: Data from Figure 7A was performed using MEFs, and are now shown in **Figure EV3B-C**, along with quantification and statistical analyses. We have included several additional experiments in primary cell types in the revised manuscript to strengthen our observations.

Figure 8. Please use plaque assays to assess viral replication. Also, the siRNA experiments lack siRNA controls.

Response: We now show MTP mediated effect using ATG5 and ATG7 CRISPR ko Neuro2a cells with plaque assay data (**Fig 9A-F**). The siRNA data contains siNT controls as indicated in **Figure 10, Appendix Fig S10**.

Text issues:

1. The authors need to define all protein acronyms and introduce their function for readers.
All protein acronyms have been defined.
2. Discussion of dopaminergic agonists is not relevant as these are an entirely different class of drugs.
Discussion of dopaminergic agonists has been reduced.

17th Nov 2023

Dear Dr. Kalia,

Thank you for the submission of your revised manuscript to EMBO Molecular Medicine. I am pleased to inform you that we will be able to accept your manuscript pending the following final amendments:

- 1) Figures: I would like to suggest substantial restructuring of the figures, so that the number of main figures is max. 8, EV figures max. 5 and the rest of the figures should be presented in the Appendix.
- 2) Please check and submit the complete Author Checklist.
- 3) In the main manuscript file, please do the following:
 - Please address all comments suggested by our data editors listed below:
 - o Please note that the figure legend style does not comply with the journal guidelines i.e. all the figure legends are in a run-on style.
 - o Please note that a separate 'Data Information' section is required in the legends of all the figures.
 - Limit keywords to max. 5.
 - Please check that all figures are correctly called out, Currently, Figure 6I is missing but it's called out in the text.
 - In M&M, provide the antibody dilutions that were used for each antibody.
 - In M&M, please specify the biosafety level for the experiments with JEV by adding and amending the following sentence: All experiments with JEV were performed in a ... level laboratory and with approval from...
 - In M&M, statistical paragraph should reflect all information that you have filled in the Authors Checklist, especially regarding randomization, blinding, replication.
 - Please rename "Disclosure Statement" to "Disclosure Statement & Competing Interests". We updated our journal's competing interests policy in January 2022 and request authors to consider both actual and perceived competing interests. Please review the policy <https://www.embopress.org/competing-interests> and update your competing interests if necessary.
 - Please correct the reference citation in the text and reference list. In the text of the manuscript, a reference should be cited by author and year of publication. Include a space between a word and the opening parenthesis of the reference that follows. In the reference list, citations should be listed in alphabetical order. Where there are more than 10 authors on a paper, 10 will be listed, followed by "et al.". Please check "Author Guidelines" for more information.
<https://www.embopress.org/page/journal/17574684/authorguide#referencesformat>
- 4) Funding: Please make sure that information about all sources of funding are complete in both our submission system and in the manuscript. Currently DBT grant BT/PR5480/INF/158/2012, SERB-National post-doctoral fellowship (PDF/2021/001436) and UGC fellowship are missing in our submission system. Also, please merge "Funding" with "Acknowledgments".
- 5) Appendix: Please correct nomenclature to "Appendix Figure S1" etc and "Appendix Table S1" etc.
- 6) The Paper Explained: Please it to the main manuscript text in the following format:
Problem:

Results:

Impact:

Please check "Author Guidelines" for more information.

<https://www.embopress.org/page/journal/17574684/authorguide#researcharticleguide>

7) Synopsis: Every published paper now includes a 'Synopsis' to further enhance discoverability. Synopses are displayed on the journal webpage and are freely accessible to all readers. They include separate synopsis image and synopsis text.

- Synopsis image: Please provide a striking image or visual abstract as a high-resolution jpeg file 550 px-wide x (250-400)-px high to illustrate your article.
- Synopsis text: Please provide a short standfirst (maximum of 300 characters, including space) as well as 2-5 one sentence bullet points that summarise the paper as a .doc file. Please write the bullet points to summarise the key NEW findings. They should be designed to be complementary to the abstract - i.e. not repeat the same text. We encourage inclusion of key acronyms and quantitative information (maximum of 30 words / bullet point). Please use the passive voice.
- Please check your synopsis text and image before submission with your revised manuscript. Please be aware that in the proof stage minor corrections only are allowed (e.g., typos).

8) For more information: This space should be used to list relevant web links for further consultation by our readers. Could you identify some relevant ones and provide such information as well? Some examples are patient associations, relevant databases, OMIM/proteins/genes links, author's websites, etc...

9) As part of the EMBO Publications transparent editorial process initiative (see our Editorial at <http://embomolmed.embopress.org/content/2/9/329>), EMBO Molecular Medicine will publish online a Review Process File (RPF) to accompany accepted manuscripts. This file will be published in conjunction with your paper and will include the anonymous referee reports, your point-by-point response and all pertinent correspondence relating to the manuscript. Let us know whether you agree with the publication of the RPF and as here, if you want to remove or not any figures from it prior to publication. Please note that the Authors checklist will be published at the end of the RPF.

10) Please provide a point-by-point letter INCLUDING my comments as well as the reviewer's reports and your detailed responses (as Word file).

I look forward to reading a new revised version of your manuscript as soon as possible.

Yours sincerely,

Zeljko Durdevic

*** Instructions to submit your revised manuscript ***

1) a .docx formatted version of the manuscript text (including Figure legends and tables)

2) Separate figure files*

3) supplemental information as Expanded View and/or Appendix. Please carefully check the authors guidelines for formatting Expanded view and Appendix figures and tables at <https://www.embopress.org/page/journal/17574684/authorguide#expandedview>

4) a letter INCLUDING the reviewer's reports and your detailed responses to their comments (as Word file).

5) The paper explained: EMBO Molecular Medicine articles are accompanied by a summary of the articles to emphasize the major findings in the paper and their medical implications for the non-specialist reader. Please provide a draft summary of your article highlighting

This may be edited to ensure that readers understand the significance and context of the research.

Please refer to any of our published articles for an example.

6) For more information: There is space at the end of each article to list relevant web links for further consultation by our readers. Could you identify some relevant ones and provide such information as well? Some examples are patient associations, relevant databases, OMIM/proteins/genes links, author's websites, etc...

7) Author contributions: the contribution of every author must be detailed in a separate section.

8) EMBO Molecular Medicine now requires a complete author checklist (<https://www.embopress.org/page/journal/17574684/authorguide>) to be submitted with all revised manuscripts. Please use the checklist as guideline for the sort of information we need WITHIN the manuscript. The checklist should only be filled with page

numbers were the information can be found. This is particularly important for animal reporting, antibody dilutions (missing) and exact values and n that should be indicated instead of a range.

9) Every published paper now includes a 'Synopsis' to further enhance discoverability. Synopses are displayed on the journal webpage and are freely accessible to all readers. They include a short stand first (maximum of 300 characters, including space) as well as 2-5 one sentence bullet points that summarise the paper. Please write the bullet points to summarise the key NEW findings. They should be designed to be complementary to the abstract - i.e. not repeat the same text. We encourage inclusion of key acronyms and quantitative information (maximum of 30 words / bullet point). Please use the passive voice. Please attach these in a separate file or send them by email, we will incorporate them accordingly.

You are also welcome to suggest a striking image or visual abstract to illustrate your article. If you do please provide a jpeg file 550 px-wide x 300-800px high.

10) A Conflict of Interest statement should be provided in the main text

11) Please note that we now mandate that all corresponding authors list an ORCID digital identifier. This takes <90 seconds to complete. We encourage all authors to supply an ORCID identifier, which will be linked to their name for unambiguous name identification.

Currently, our records indicate that the ORCID for your account is 0000-0002-3376-3659.

Please click the link below to modify this ORCID:
Link Not Available

Graphs 800-1,200 DPI
Photos 400-800 DPI
Colour (only CMYK) 300-400 DPI"

*Additional important information regarding figures and illustrations can be found at
<https://bit.ly/EMBOPressFigurePreparationGuideline>. See also figure legend preparation guidelines:
<https://www.embopress.org/page/journal/17574684/authorguide#figureformat>

***** Reviewer's comments *****

Referee #2 (Remarks for Author):

The authors have adequately addressed my critiques and concerns.

The authors address the minor editorial issues.

24th Nov 2023

Dear Dr. Kalia,

We are pleased to inform you that your manuscript is accepted for publication and is now being sent to our publisher to be included in the next available issue of EMBO Molecular Medicine.
